# Evaluation of ORCHIDEE-MICT simulated soil moisture over China and impacts of different atmospheric forcing data

Zun Yin[1], Catherine Ottlé[1], Philippe Ciais[1], Matthieu Guimberteau[1,2], Xuhui Wang[1,3,4], Dan Zhu[1], Fabienne Maignan[1], Shushi Peng[3], Shilong Piao[3], Jan Polcher[4], Feng Zhou[3], Hyungjun Kim[5], and other China-Trend-Stream project members[*]

[1]Laboratoire des Sciences du Climat et de l'Environnement, CNRS-CEA-UVSQ, Gif-sur-Yvette 91191, France
[2]UMR 7619 METIS, Sorbonne Universités, UPMC, CNRS, EPHE, 4 place Jussieu, Paris 75005, France
[3]Sino-French Institute for Earth System Science, College of Urban and Environmental Sciences, Peking University, Beijing 100871, China
[4]Laboratoire de Météorologie Dynamique, UPMC/CNRS, IPSL, Paris 75005, France
[5]Institute of Industrial Science, The University of Tokyo, Tokyo, Japan
[*]A full list of the China-Trend-Stream project members and their affiliations appears at the end of the paper.

*Correspondence to:* vyin@lsce.ipsl.fr

**Abstract.** Soil moisture is a key variable of land surface hydrology and its correct representation in land surface models is crucial for local to global climate predictions. The errors may come from the model itself (structure and parameterization) but also from the meteorological forcing used. In order to separate the two source of errors, four atmospheric forcing datasets: GSWP3 (Global Soil Wetness Project Phase 3), PGF (Princeton Global meteorological Forcing), CRU-NCEP (Climatic Research Unit-National Center for Environmental Prediction), and WFDEI (WATCH Forcing Data methodology applied to ERA-Interim reanalysis data), were used to drive simulations in China by the land surface model ORCHIDEE-MICT. Simulated soil moisture was compared with in-situ and satellite datasets at different spatial and temporal scales in order to: 1) estimate the ability of ORCHIDEE-MICT (ORganizing Carbon and Hydrology in Dynamic EcosystEms: aMeliorated Interactions between Carbon and Temperature) to represent soil moisture dynamics in China; 2) demonstrate the most suitable forcing dataset for further hydrological studies in Yangtze and Yellow river basins; 3) understand the discrepancies of simulated soil moisture among simulations. Results showed that ORCHIDEE-MICT can simulate reasonable soil moisture dynamics in China, but the quality varies with forcing data. Simulated soil moisture driven by GSWP3 and WFDEI shows the best performance according to RMSE and correlation coefficient respectively, suggesting that both GSWP3 and WFDEI are good choices for further hydrological studies in the two catchments. The mismatch between simulated and observed soil moisture is mainly explained by the bias of magnitude, suggesting that the parameterization in ORCHIDEE-MICT should be revised for further simulations in China. Underestimated soil moisture in the North China Plain demonstrates possible significant impacts of human activities like irrigation on soil moisture variation, which was not considered in our simulations. Finally, the discrepancies of meteorological variables and simulated soil moisture among the four simulations are analyzed. The result shows that the discrepancy of soil moisture is mainly explained by differences in precipitation frequency and air humidity rather than differences in precipitation amount.

## 1 Introduction

Climate change strongly influences the hydrological cycle, which in turn affects ecosystems services, food security, and water resources (Bonan, 2008; Piao et al., 2010; Seneviratne et al., 2010; Zhu et al., 2016). More importantly, the main mechanisms
governing hydrological process vary across climate regimes under anthropogenic factors (Guimberteau et al., 2012; Wada et al., 2016, 2017). Covering different climate zones and most types of human activities (An et al., 2017; Basheer and Elagib, 2018; Bouwer et al., 2009; Feng et al., 2016; Rogers et al., 2016; Wu et al., 2018), China is a good test bed to investigate the hydrological complexity of climate-water-human interactions. In China, annual precipitation increased in the South but declined in the North over the last several decades (Ye et al., 2013; Zhai et al., 2005). This dipole of precipitation trends is partly
reflected in the discharge trends of Yangtze and Yellow rivers (Piao et al., 2010), but other factors than precipitation changes affect river discharge including changes in rainfall intensity, land surface state or condition, and water management (Ayalew et al., 2014; Grillakis et al., 2016; Williams et al., 2015). A prerequisite to understand how precipitation changes transfer into river discharge changes is to analyze and evaluate the various components of the surface water budget and especially the key variable relationships between precipitation and soil moisture (SM), result of the partition of precipitation among
evapotranspiration, infiltration, and runoff.

SM indeed plays a crucial role in adjusting local climate (Seneviratne et al., 2013; Teuling et al., 2010), regulating productivity, and ecosystem dynamics (Schymanski et al., 2008; Yin et al., 2014) and affecting carbon budgets (Calvet et al., 2004). SM controls vegetation photosynthesis through transpiration, which in turn significantly influences surface temperature (Bonan, 2008; Dai et al., 2004). It also impacts the infiltration rate of precipitation in the soil and its state before rainfall events
determines the ratio of surface runoff to precipitation (Grillakis et al., 2016). Therefore SM is not only of importance in understanding land surface processes, but also is a key indicator for predicting and addressing extreme events, such as heat waves, floods, and droughts (Hirschi et al., 2011; Teuling et al., 2010; Wanders et al., 2014).

To investigate the spatial and temporal SM variations, in-situ measurements (Dorigo et al., 2011; Liu et al., 2001; Piao et al., 2009; Robock et al., 2000) are too sparse and not always representative of larger scales. Although they can provide first hand
records of SM fluctuations, the density of in-situ networks cannot meet the requirement for continental scale studies. And the different measurement techniques make it difficult to combine different datasets. Satellite-based SM products (Dorigo et al., 2015; Njoku et al., 2003; Su et al., 2003; Wagner et al., 2012) provide excellent spatial coverage and temporal sampling, but their accuracy varies between instruments and retrieval algorithms used (Liu et al., 2012). Moreover, these estimations concern only the first centimeters of soil but root zone SM cannot be directly assessed, unless a model simulating the water
transfer processes is used. To overcome the uneven coverage of raw data, data assimilation is widely applied to analyze SM from in-situ or satellite observations (Draper et al., 2012; Martens et al., 2016; Reichle et al., 2007). Analyzed products help us understanding SM variation and its relation to climate (Liu et al., 2015b, 2017; Taylor et al., 2012). However, to capture changes of hydrological mechanisms for future projection, such measurements are not enough.

Land surface models (LSMs) are able to simulate the short- and long-term SM dynamics consistently with atmospheric forcing and surface information (Pierdicca et al., 2015; Rebel et al., 2012; Xia et al., 2014) by reproducing physical processes, and interactions with other climatic, hydrological and ecological factors (Seneviratne et al., 2010). The uncertainty of simulated SM depends on the accuracy of atmospheric forcing, in particular precipitation frequency and intensity, and radiation. However LSMs complexity is a source of structural errors (missing processes) and biased parameters. Thus it is necessary to validate simulated SM by observations in order to diagnose the source of errors and estimate the ability of the chosen LSM to simulate SM dynamics in the area of interest.

In this study, the land surface model: ORCHIDEE-MICT (ORganizing Carbon and Hydrology in Dynamic EcosystEms: aMeliorated Interactions between Carbon and Temperature; Guimberteau et al. (2018)) is used to simulate SM over China. Besides land surface hydrology, ORCHIDEE-MICT simulates energy budgets and vegetation dynamics (mechanistic phenology, photosynthesis, and ecosystem carbon cycling), which interact with the water cycle and climate (Guimberteau et al., 2012). Moreover, the evaluation of simulated SM controlled by natural processes is useful to identify human effects (e.g., crops, irrigation, dam operation, etc) on water budget in regions where there is a large misfit between model and observation.

Four global atmospheric forcing datasets are chosen to drive the simulations in China, including GSWP3 (Global Soil Wetness Project Phase 3), PGF (Princeton Global meteorological Forcing), CRU-NCEP (Climatic Research Unit-National Center for Environmental Prediction), and WFDEI (WATCH Forcing Data methodology applied to ERA-Interim reanalysis data), due to their widely applications in numerous hydrological studies (Getirana et al., 2014; Guimberteau et al., 2014, 2017, 2018; Hirschi et al., 2014; Van Den Hurk et al., 2016; Polcher et al., 2016; Schmied et al., 2016; Tangdamrongsub et al., 2018; Yang et al., 2015; Zhao et al., 2017; Zhou et al., 2018). Although they provide gridded surface climate variables at global scale, their uncertainties of representing regional climate are not clear. Through comparison of simulated SM to various datasets, our study also addresses which forcing has the best performance in SM simulation in China.

Our SM simulations are evaluated with different SM datasets including in-situ, remote sensing measurements, and reanalysis. In-situ measurements including ISMN (International Soil Moisture Network; Dorigo et al. (2011)) and PKU (in-situ SM from Peking University; Piao et al. (2009); Xu (2014)) are used to evaluate temporal validation of simulated SM. To evaluate spatio-temporal variations of simulated SM, the satellite based dataset ESA CCI SM (European Space Agency Climate Change Initiative Soil Moisture; Wagner et al. (2012)) is applied in the comparison. Note that both in-situ and satellite SM datasets represent the 'truth' to some extent. This implies that real-world SM is influenced by processes that are not modeled such as irrigation and wetlands. Thus mismatches between measured and simulated SM may exist in some regions strongly affected by anthropogenic factors. Moreover, satellite instruments do not measure directly SM which is derived via a complex modelization of the radiative transfer at the soil-vegetation interface calibrated with in-situ data.

Finally the GLEAM SM data (The Global Land Evaporation Amsterdam Model; Martens et al. (2017)) is compared to the simulated SM. Different from other SM datasets, GLEAM SM results from a land surface model constrained by a number of satellite and in-situ observations. This reanalysis product was shown to reproduce reasonable long period SM dynamics at global scale (Martens et al., 2017), which is valuable to evaluate ORCHIDEE-MICT simulations for both surface and root-zone

SM. Furthermore, GLEAM assimilates CCI SM data, so that evaluation of our model against root-zone SM from GLEAM is consistent with evaluation against surface SM from CCI.

Through the simulations and comparisons, three questions will be addressed:

- Is the model able to provide a reasonable estimation of SM dynamics in China, as a prerequisite for further hydrological studies?

- Which atmospheric forcing gives the best SM simulation according to the comparisons with available observations?

- Which meteorological variable drives the differences of SM among the simulations?

The study area, atmospheric forcing, and SM datasets used in this study are described in Section 2. Section 3 presents the model experiments. Evaluation of simulated SM and discussion are given in Section 4 and 5 respectively.

## 2 Study area, forcing, and evaluation datasets

### 2.1 Study area

China has multiple climate regimes, which makes hydrological situations influenced by different variables in different regions. The land water budget in China is affected by anthropogenic factors, such as irrigation (Puma and Cook, 2010), afforestation (Liu et al., 2015a; Peng et al., 2014), deforestation (Wei et al., 2018), polders (Yan et al., 2016), dams (Deng et al., 2016), and inter-basin water transfer (Li et al., 2015). Two main river basins are of interest: the Yangtze River Basin (YZRB) and the Yellow River Basin (YLRB) (red and magenta contours respectively in Fig. 1), which cover the main regions of industry and agriculture (gray regions in Fig. 1). The Yangtze River originates in the Qinghai-Tibetan Plateau and flows through two wetted traditional agricultural zones: Sichuan Basin and the plain at the downstream of the Yangtze River (Fig. 1). The Yellow River originates in the Qinghai-Tibetan Plateau as well, but it flows through another two agricultural regions (the Loess Plateau and the North China Plain) under semi-arid and semi-humid zones (Kottek et al., 2006). Our simulations cover the main part of China ([85-124°E]×[20-44°N]) including these two watersheds to assess SM dynamics at catchment scale. Note that in the analysis, the specific regions of the two river basins are coarser than the exact basin contours shown in Fig. 1 due to the interpolation of routing files at the resolution of our simulations.

### 2.2 Atmospheric forcing

Four gridded atmospheric forcing datasets are used to force the model over China: GSWP3, PGF, CRU-NCEP, and WFDEI. All input variables needed are the air temperature at 2 m ($T_a$), rainfall and snowfall rates, atmospheric specific humidity at 2 m ($Q_a$), surface pressure, downward short/long wave radiation ($R_s$ and $R_l$), and wind speed ($W$). The four forcing datasets are combinations of reanalysis and observation data. These datasets, although built by different methods, are not independent from each other since they share some common inputs. Detailed descriptions are listed below and general information is summarized in Table 1. Preprocessing of the datasets for ORCHIDEE-MICT is described in Sect. 3.3.

**GSWP3**

The GSWP3 v0 (http://hydro.iis.u-tokyo.ac.jp/GSWP3); Kim (2017)) provides a 3-hourly climate data at 0.5° resolution from 1901 to 2010. It is based on the 20th Century Reanalysis (20CR; Compo et al. (2011)), which is downscaled from 2° to 0.5° by a spectral nudging technique in a Global Spectral Model (Yoshimura and Kanamitsu, 2008), in order to maintain both low and high frequency signals at high spatio-temporal scale. Single ensemble correction and vertically weighted damping are applied to remove known artifacts in high latitude regions (Hong and Chang, 2012; Yoshimura and Kanamitsu, 2013). Moreover, observation data are used for bias-correction, such as GPCC v6 (Global Precipitation Climatology Centre; Becker et al. (2013)) for precipitation, SRB (Surface Radiation Budget; Stephens et al. (2012)) for radiation and CRU TS v3.21 (Climate Research Unit; Harris et al. (2014)) for temperature.

**PGF**

The PGF (http://hydrology.princeton.edu/data.pgf.php, latest version released on 13 Jul, 2014) provides 3-hourly data at 1° resolution from 1901 to 2012 (Sheffield et al., 2006). It is constructed by combining the NCEP-NCAR (National Centers for Environmental Prediction-National Center for Atmospheric Research) reanalysis of Kalnay et al. (1996) with several observation datasets. Precipitation is corrected by downscaled CRU TS v3.1, GPCP (Global Precipitation Climatology Project; Adler et al. (2003)), and TRMM (Tropical Rainfall Measuring Mission; Huffman et al. (2007)) data. SRB and CRU TS data are used in the assimilation of radiation and air temperature, respectively. Other variables (e.g., specific humidity, surface air pressure, wind speed) are just spatially downscaled from NCEP-NCAR according to the local elevation.

**CRU-NCEP**

The CRU-NCEP v6.1 (ftp://nacp.ornl.gov/synthesis/2009/frescati/modeldriver/cru ncep/analysis/readme.htm) provides 6-hourly 0.5° data. It combines the coarse temporal resolution (monthly) CRU TS dataset with the NCEP reanalysis, which has a higher time interval (6-hourly) but is only available at 2.5°. Monthly climate (except for precipitation) is identical to CRU TS, and NCEP is used only to reconstruct the 6-hourly variability within each month after bi-linearly interpolated to 0.5°. For precipitation the original NCEP values are used for temporal linear interpolation in those CRU grid cells (0.5°) covered by the specific NCEP grid cell (2.5°) in each month. CRU-NCEP dataset is available from 1901 to 2015 at global scale and it is updated every year.

**WFDEI**

The WFDEI forcing (version 31 Jul 2012) is generated by applying the WATCH Forcing Data methodology (http://www.eu-watch.org, Weedon et al. (2014)) to the ERA-Interim reanalysis (Dee et al., 2011) providing 3-hourly data at 0.5° from 1979 to 2009. The ERA-Interim blends GCM modeled variables and a suite of observations by a 4D-Var (4-dimensional variable analysis) data assimilation system (Weedon et al., 2014). All variables are bias-corrected using CRU TS. For precipitation, we use a version that has been bias-corrected by GPCC v5 and v6.

## 2.3 Soil moisture datasets

**International Soil Moisture Network ISMN**

ISMN is an international cooperative project providing a global gauged SM database (Dorigo et al., 2011). It is based on in-situ measurements from multiple monitoring regional sub-projects. Here only data from the CHINA sub-project is used (Robock et al., 2000) with in-situ volumetric water content (depth of water column over depth of soil in $m^3.m^{-3}$) from 40 stations between 1981 and 1999. SM profiles on 11 vertical layers were collected three times per month and (on 8th, 18th, and 28th of each month). The 11 sampled soil layers are: 0-5 cm, 5-10 cm, and then every 10 cm layers until 1 m. Most stations are located in cropland or grasslands, but information about land use types and soil texture of each site are not provided. Moreover, there is no information about management practices affecting SM, such as irrigation or tillage.

In spite of the long length of this dataset, the data availability and monitoring period among stations vary widely. Some stations only recorded SM during the growing season, while others have a full year record. Furthermore, the measurements including the 5 deep layers (below 50 cm) are less than those including the top 6 layers. Only stations with more than 15 years of data were selected, which at least cover the same period (1984-1999). To make sure that there is at least half of the data available in the 15-year time series, stations with less than 270 measurement points in the top 6 layers are removed. This leads to selecting a subset of 20 stations, and given the sparseness of data below 50 cm, only SM in the top 6 layers is used for model evaluation.

**In-situ SM from Peking University**

The SM was measured over 778 stations of agro-meteorological stations over China by the Chinese Meteorological Administration (Xu, 2014) and collected and harmonized by the research team in Peking University (PKU; Piao et al. (2009)). The dataset provides 10-day SM variation during the growing season (mainly between May and September) from 1991 to 2007. It provides SM profiles in 7 soil layers (0-10 cm, 10-20 cm, 20-30 cm, 30-40 cm, 40-50 cm, 50-70 cm, and 70-100 cm), but the bottom four layers often have missing records. This dataset concerns exclusively croplands but there is no explicit information of soil texture and irrigation. Similar to the ISMN, the monitoring durations among gauging stations are different. 203 stations that cover the period of 1992-2006 are chosen.

**ESA CCI SM**

The ESA CCI SM is a multi-satellite based product (Liu et al., 2011, 2012; Wagner et al., 2012) and has been validated both at global scale (Dorigo et al., 2015) and in China (Peng et al., 2015; An et al., 2016). The daily SM is retrieved from a suite of microwave sensors spreading the period of 1979-2010 with 0.5° resolution. The representative soil layer depth is approximately 0.5-2 cm. Multi sensors ensure a long term records of SM dynamics, however the uncertainty of the data varies with the change of available sensors and corresponding algorithms. Moreover, the remote sensing technique limits its ability to detect SM in frozen soils or under snow cover. Therefore SM data is not available during winter in high latitude regions (e.g., Northern

China). The data availability also varies along the period according to the number of available instruments and the increase of their temporal and spatial resolutions. In China, the fraction of days with available records (Figure 4 of Dorigo et al. (2015)) is lower than 20% from 1979 to 2006. More importantly, large spatial variation of gaps exists as well before 2006 (Fig. B1). The period after the launch of METOP-A-ASCAT (Advanced Scatterometer) at the end of 2006 appears much stable. To provide a

reliable validation, we only use the CCI SM data between 2007 and 2009.

**GLEAM v3.0A SM**

The GLEAM v3.0 is a multiple algorithm, observation-based model reconstructing the components of the land evaporation process, including daily SM, evapotranspiration, and interception at $0.25°$ resolution (Martens et al., 2017). It has three sub versions. Due to the short duration of version B (2003-2015) and C (2011-2015) only version A, which covers the period

1980-2014, is used here. Radiation and air temperature used in GLEAM 3.0A are from ERA-Interim, and precipitation is from MSWEP (Multi-Source Weighted-Ensemble Precipitation; Beck et al. (2016)).

Both surface and root-zone SM from GLEAM, which has been validated by Martens et al. (2017), are used for comparison. The surface SM in the top 0-10 cm is a combination of simulated SM from the GLEAM soil module, SMOS (the Soil Moisture Ocean Salinity satellite mission; Kerr et al. (2001)), and ESA CCI SM (ESA Climate Change Initiative Soil Moisture; Liu et al.

(2011, 2012); Wagner et al. (2012)) through the data assimilation system developed by Martens et al. (2016). The Community Noah land surface model SM fields in GLDAS (Global Land Data Assimilation System; Rodell et al. (2004)) was used to estimate the errors of these SM products. Root-zone SM is derived from the GLEAM soil module based on mass balance. GLEAM provides SM in separate land-cover tiles of bare soil (0-10 cm), low vegetation (0-100 cm), and tall vegetation (0-250 cm). These tiles are based on MODIS Vegetation Continuous Fields (MOD44B; Hansen et al. (2003)).

**3   Land surface model, simulation protocol, and model-data comparison metrics**

**3.1   Land surface model**

ORCHIDEE (Organizing Carbon and Hydrology In Dynamic EcosystEms; Krinner et al. (2005)) is a physical-based land surface process model. It is mainly composed by two modules. The SECHIBA (surface-vegetation-atmosphere transfer scheme) module calculates the exchange of water and energy between land and atmosphere with a high time interval (half an hour).

While the STOMATE (Saclay Toulouse Orsay Model for the Analysis of Terrestrial Ecosystems) module estimates the carbon cycle at daily time scale. The ORCHIDEE-MICT (aMeliorated Interactions between Carbon and Temperature, SVN version 3952; Guimberteau et al. (2018); Zhu et al. (2015)) is a recent version of ORCHIDEE including new processes as the interactions among frozen soil, snow, plants, and soil carbon pools. It accounts for soil freezing, soil carbon discretization, snow processes, and lateral water flows to improve the simulation of the main biogeochemical cycles in permafrost regions. It has

been chosen in this study because China has a large permafrost area especially for the Tibetan Plateau, where originate both Yellow and Yangtze rivers. To simulate the SM dynamics, ORCHIDEE-MICT uses a 11-soil layer scheme, whose depth in-

creases exponentially until 2 m. The respective depths (in meters) of the calculation nodes are the following: 0.0005, 0.002, 0.006, 0.014, 0.03, 0.06, 0.12, 0.25, 0.5, 1.0, and 1.75. Each grid cell can include up to three soil tiles: bare soil, trees, and grass/crops, which are filled by the corresponding plant functional types (PFT) of the 13-PFT scheme of ORCHIDEE-MICT to allow better representation of their specific hydrology. The hydrological budget is calculated separately in each soil tile.

The amplitude of SM depends on soil texture, which is a part of boundary conditions. Explicit description of the ORCHIDEE-MICT model can be found in Guimberteau et al. (2018). ORCHIDEE will be referred to ORCHIDEE-MICT for brevity in the following text.

There are two main outputs of SM in ORCHIDEE. The total SM ($\theta_t$) indicates the total amount of soil water volume in the top 2 m soil layer in a grid cell. The SM profile ($\theta_p$) records the vertical distribution of soil water content in the 11 soil layers.

Note that the $\theta_p$ in each soil layer is an average value among the three soil tiles. The initial unit of ORCHIDEE SM is $m^3.m^{-3}$.

## 3.2   Simulation protocol

Four simulations were performed driven by different forcing datasets described in Sect 2.2. In the simulations, $CO_2$ rise, and land use change are taken into account but without human processes like irrigation. The 13-PFT map is from LUH2 (http://luh.umd.edu) and the soil texture map is from Zobler (1986). For the 3-soil texture scheme of Zobler86, the minimum

residual and maximum saturated SM are 0.065 and 0.43 $m^3.m^{-3}$, respectively. The model domain covers the main part of China ([85-124°E]×[20-44°N]). The spatial resolution is as same as the atmospheric forcing (Table 1). The simulation period covers 39 years, from 1971 to 2009, except for the one driven by WFDEI, which is from 1979 to 2009. To make sure that carbon (LAI and biomass) and water cycle variables can reach equilibrium, a 100-year spin-up was performed by repeating 10 times the forcing of the period 1971-1980 (for WFDEI, 50 times the period 1979-1980). Starting from the end of the spin-

up, simulations were run from 1981 to 2009. The output driven by PGF forcing was re-gridded at $0.5° \times 0.5°$ to match the resolution of other simulation outputs.

The temporal resolution of forcing datasets is either 3-hourly or 6-hourly (CRU-NCEP), which is larger than the simulation time step of SECHIBA (30 min). To have a reasonable precipitation intensity and thus a good infiltration of water in the soil, the default precipitation splitting algorithm of ORCHIDEE is applied in our simulations. At the beginning of each forcing time

step, if precipitation occurred, the precipitation amount (precipitation rate multiplied by the time interval of specific forcing) will be uniformly distributed to the first half of the forcing time step.

## 3.3   Model-data comparison methodology and metrics

**Comparison protocol**

As the soil depths, periods, and spatio-temporal resolutions are different in the four SM datasets (Sect. 2.3), we have to chose

corresponding ORCHIDEE outputs for each comparison. To compare with the in-situ data of ISMN and PKU, we first extracted modeled daily SM profile ($\theta_p$) from the nearest grid cell for each station. Then the SM above a certain soil depth was chosen (50 cm for ISMN and 20 cm for PKU). PKU SM is provided in degree of saturation, defined as the volume ratio of actual

water content to its maximum value when the soil is saturated. As the soil porosity is unknown, the PKU SM dataset cannot be directly compared with simulated SM from ORCHIDEE, which is defined from modeled porosity. To overcome this problem, normalization was applied on both datasets before comparison. The normalized data at each station and in the corresponding grid cell of the model is the ratio of the difference between the original value and its mean (during the observation period) to its standard deviation.

According to the sampled depth of the ESA CCI SM, the daily top 4-layer (2.2 cm) averaged SM from ORCHIDEE is used. Regarding the definition of GLEAM SM (Sect. 2.3), we used the daily top 6-layer (approximately 9.2 cm depth) averaged SM and the total SM of ORCHIDEE to compare with GLEAM surface and root-zone SM, respectively. The period length and soil depth of each comparison are shown in Table 2. In addition, the timing of all SM datasets is uniformed to the Coordinated Universal Time (UTC).

## Metrics

Pearson correlation coefficient ($r$) is calculated to estimate the correlation between simulated and observed SM. Daily SM corresponding to the measurement date reported in ISMN was collected to calculate $r$. As there is no date information from the 10-day PKU dataset, we used the 10-day averaged SM from ORCHIDEE for comparison.

The Root Mean Square Error (RMSE) is applied in order to estimate the temporal differences between simulation and observation. The same data pairs are used for RMSE calculation as the correlation coefficient except for PKU due to the normalization. Note that RMSE is related to the magnitude of SM, which varies significantly in China. To make it comparable in space, relative RMSE is calculated by dividing the mean of the simulated and observed SM.

According to Kobayashi and Salam (2000), the mean squared deviation (MSD), which is $\text{RMSE}^2$, can be decomposed into squared bias (SB), squared difference between standard deviation (SDSD), and lack of correlation weighted by the standard deviation (LCS), as:

$$\text{MSD} = \text{RMSE}^2 = \text{SB} + \text{SDSD} + \text{LCS}. \tag{1}$$

SB is the bias between simulations and observations. It is independent from other two components:

$$\text{SB} = (\bar{s} - \bar{m})^2, \tag{2}$$

where $\bar{s}$ and $\bar{m}$ are the mean of simulated and measured values, respectively. The SDSD indicates the mismatch of variation magnitude between simulated and observed variables, defined as:

$$\text{SDSD} = (\text{SD}_s - \text{SD}_m)^2, \tag{3}$$

where $\text{SD}_s$ and $\text{SD}_m$ are standard deviation of simulations and measurements, respectively. High SDSD implies a failure of the model in simulating the degree of fluctuation across the $n$ measurements. Note that SDSD correlates with LCS, which accounts for $\text{SD}_s$ and $\text{SD}_m$ as well:

$$\text{LCS} = 2\text{SD}_s\text{SD}_m(1 - r), \tag{4}$$

where $r$ is the Pearson correlation coefficient. The LCS is an indicator of the performance of the model to simulate the pattern of fluctuation of the measurements. The lower the LCS is, the better the model performs.

Finally, to evaluate the characteristic time scale of modeled SM response to hydrological processes, lag-$k$ autocorrelation coefficient ($R_k$) is computed. The $R_k$ is the correlation coefficient of a time series with itself but with a $k$ time step lag, as:

$$R_k = \frac{\sum_{i=1}^{n-k}(x_i - \bar{x})(x_{i+k} - \bar{x})}{\sum_{i=1}^{n}(x_i - \bar{x})^2}, \tag{5}$$

where $n$ ($n > k$) is the length of the specific time series; $x$ is the mean value. For SM time series in a specific grid cell, $R_k$ was computed for different $k$ values. The value of $R_k$ decreases with increasing $k$ and the $k$-lag time series are considered not auto-correlated if $R_k$ is less than a threshold $1/e$ (Maurer et al., 2001; Rebel et al., 2012). The day number when $R_k$ first drops below a threshold of $1/e$ is called number of lag days (NLD).The NLD difference is used to compare the overall characteristic time scales between datasets. The difference of $R_k$ profiles gives additional information on the autocorrelations for lag. The $R_k$ comparison was implemented between GLEAM and ORCHIDEE because other datasets do not have complete daily records.

The linear trend of SM change in the 29 years is of interest as well. The Mann-Kendall test (Kendall, 1975; Mann, 1945) is applied to test if simulations capture observed trends of SM, with $p$-value $< 0.05$ indicating a significant trend.

### 3.4 Correlation of uncertainties between SM and meteorological factors

In our simulations, the difference in atmospheric forcing is the only source of difference in simulated SM. We look at different climate variables to explain SM differences among simulations. These variables include monthly precipitation amount ($P$) and the number of precipitation days in one month ($N_p$) excluding days with $P < 0.01$ mm.d$^{-1}$. Precipitation days are categorized into 5 classes of 0.01-1, 1-5, 5-10, 10-15, and $> 15$ mm.d$^{-1}$. The number of days with precipitation amount in each class was calculated, denoted by $N_p^i$ with $1 \leq i \leq 5$. Other meteorological variables are incoming short/long wave radiation ($R_s/R_l$), air temperature ($T_a$), air humidity ($Q_a$), and wind speed ($W$). Regarding SM, both total SM ($\theta_t$) and SM in each soil layer ($\theta_p^i$, $i$ is the index of soil layer) were correlated with these variables. To estimate the difference of a variable x among the four simulations, the averaged MSD ($D_x$) is computed as:

$$D_x = \frac{\frac{1}{n}\sum_{i \neq j}^{N}\sum_{t=1}^{n}(x_{t,i} - x_{t,j})^2}{\binom{n}{2}}, \tag{6}$$

where $N = 4$ is the number of simulations; $i$ and $j$ ($1 \leq i,j \leq N$) are indexes of the four simulations; $\binom{n}{2}$ is the binomial coefficient; $n$ is the length of the time series; $t$ is the time step. Note that we use the absolute value of $D_x$ not relative $D_x$ ($D_x$ over averaged value of $x$ in the specific grid cell) for the analysis because the relative $D_x$ cannot reflect the linkage of uncertainty between inputs and outputs. Detailed explanation is shown in Supplement C.

## 4 Results

### 4.1 SM evaluation against multiple datasets

**Comparison with ISMN and PKU in-situ data**

In most cases, the correlations between modeled and measured SM at ISMN stations (see Sect. 2.3) are significantly positive (Fig. 2). High correlations ($r > 0.6$) are found over the Loess Plateau in the semi-arid zone, where is the region of rainfed agriculture and SM is less affected by anthropogenic processes. In the North China Plain water is limited as well, whereas irrigation is widely applied for agriculture leading to low $r$ (below 0.5). To further compare the simulated and measured SM, three ISMN stations (marked by squares in Fig. 2(a)) are chosen to represent for different wet conditions and model-data comparisons are shown in Figure 3. Xifeng locates in the semi-arid zone (MAP = 556 mm.yr$^{-1}$), where $\theta_t$ is low (0.2 m$^3$.m$^{-3}$ on average) with a large inter-annual variation. The variability of simulated $\theta_t$ is consistent with observations ($0.73 < r < 0.87$; when CRU-NCEP is excluded) due to lower human impacts on rainfed agriculture in this region (Li et al., 2014). Xinxian is located in the North China Plain with similar MAP (580 mm.yr$^{-1}$) to Xifeng, but in a traditional irrigation region (Wang et al., 2016). $\theta_t$ at Xinxian is underestimated, possibly because irrigation is not included in our simulations. Thus the model cannot capture the seasonal variations of $\theta_t$, given $r$ values ranging between 0.11 and 0.21. Xuzhou is in the North China Plain as well but with a higher MAP (847 mm.yr$^{-1}$). The fluctuation of simulated and observed $\theta_t$ are coherent, leading to $r$ from 0.55 to 0.64. However the magnitude of $\theta_t$ is systematically underestimated as well (Fig. 3).

The correlation coefficients of $\theta_t$ between simulations and PKU dataset are shown in Figure 2 as circles. Modeled $\theta_t$ has a better performance in the Loess Plateau and the North China Plain than other regions, suggesting that ORCHIDEE is able to capture the variations of SM in semi-arid and temperate zones. In comparison to ISMN, $r$ between ORCHIDEE and PKU $\theta_t$ is lower. This may be caused by the shallower depth of the PKU data (20 cm) with stronger influence from fast infiltration and transpiration processes than in the ISMN records (1 m). Moreover, the PKU dataset only records $\theta_t$ during the growing season, leading to lower $r$ in absence of full seasonal variations. Negative correlations are found in several sites located along river networks. The negative $r$ (-0.4 < $r$ < -0.2) coincides with the coupling of wetness anomaly and irrigation: when droughts occur (reflected by low simulated SM), more water will be withdrawn from the river and irrigated on the crop lands (reflected by high observed SM). Thus the negative $r$ found in Fig. 2 reveals that SM dynamics cannot be well understood without considering anthropogenic activities.

According to the $r$ shown in Figure 2, we find that GSWP3 and WFDEI provide better simulated SM than the other two. The main difference is found in the North China Plain, where the $r$ values of GSWP3 and WFDEI are higher. It indicates that simulation can be improved by selecting suitable atmospheric forcing. Nevertheless, the $r$ is still limited by the lack of measurement information (soil texture, irrigation flag, land cover, etc), and anthropogenic processes in ORCHIDEE. The disagreements between simulated and measured SM are caused by the spatial scale as well. The spatial resolutions of forcings ($0.5° \approx 55$ km) are too coarse to represent the specific climatic conditions of gauging stations. On the other hand, the com-

parison cannot provide a comprehensive validation in the YZRB where few measurements locate in. Thus remote sensing and hybrid SM datasets are required to evaluate the simulations.

**Comparison with ESA CCI SM data**

Figure 4 (left panel) shows $r$ between CCI and ORCHIDEE $\theta_s$ from 2007 to 2009. High $r$ is found in both the North China Plain and the southern China. In southern China, SM is less disturbed by anthropogenic factors due to its wet condition. Thus SM has small variation driven by climate, which can be well simulated by the model. On the other hand, human activities strongly affect SM in the North China Plain, whereas their impacts on $r$ are neutralized by the large annual variation due to seasonality. Weak correlations only exist in the transition zone from the south to the north along the Yangtze river network, where compounds both human disturbances and small annual variation.

Figure B2 and 4 (right panel) show the relative RMSE and the MSD decomposition of $\theta_s$ between CCI and ORCHIDEE. Low relative RMSE ($< 0.3$) is found in the YZRB, but in the YLRB the value is higher ($> 0.4$). The mean source of MSD is LCS (phase mismatch). It implies that the magnitude of the simulated $\theta_s$ is reasonable, but the timing of the fluctuations differs between ORCHIDEE and CCI. The coincidence of magnitude is reflected from the relative difference (Fig. B3), the absolute value of which is less than 0.1 in 76% grid cells excluding the CRU-NCEP case. Large LCS might be due to human activities and the discretization of CCI SM time series. Irrigation in the northern China may significantly affect the fluctuation of $\theta_s$, which leads to underestimation of simulated SM and contributes to the LCS. Simultaneously, due to the incomplete records of CCI SM (Fig. B1) the seasonal variation of SM cannot be fully taken into account in the comparison. The $r$ is consequently declined and the LCS increases (Eq. 4).

The availability and uncertainty of CCI SM vary with space and time (Sect. 2.3 and Fig. B1). To provide reliable estimation, we performed the analysis exclusively in the period 2007-2009. In fact, there is few difference if the comparison covered the whole period 1981-2009. The patterns of $r$ and MSD decompositions (Fig. B4) are similar to that of comparison 2007-2009. The $r$ of 1981-2009 is lower with no doubt, because longer period contains more errors due to the fragmentary records of CCI SM data.

**Comparison with GLEAM v3.0A data**

The left panel of Figure 5 shows correlation coefficients between GLEAM surface SM ($\theta_s$) and corresponding modeled SM in the surface layer (0-10 cm). Simulated $\theta_s$ is significantly correlated with GLEAM (median $r = 0.54$). In the Sichuan Basin, $r$ is lower than its surroundings. According to the spatially averaged $r$ of $\theta_s$, GSWP3 (0.55) and WFDEI (0.66) lead to better performances with ORCHIDEE than PGF (0.43) and CRU-NCEP (0.51). Note that both WFDEI and GLEAM v3.0A used ERA-Interim reanalysis to reconstruct the time series of precipitation, which can explain the higher $r$ when ORCHIDEE is forced by WFDEI. The correlation coefficients of simulated and GLEAM root-zone SM ($\theta_r$, Fig. B5) have the similar patterns as the $\theta_s$ but higher values (median $r = 0.57$) due to the lower variability of $\theta_r$, which smoothes out misfits related to differences in individual rainfall events between ORCHIDEE and GLEAM for $\theta_s$. Compared to CCI $\theta_s$, the $r$ between ORCHIDEE and GLEAM $\theta_s$ is much higher. It is probably due to the shallower depth of the CCI $\theta_s$, which is more sensitive to surface processes

and forcing data errors. Moreover, CCI $\theta_s$ is a purely satellite product while GLEAM $\theta_s$ (v3.0A) is a combination of modeled, in-situ and satellite SM. The latter one is not totally independent of the forcing datasets and therefore more comparable to our simulations.

Figure B6 and the right panel of 5 show the relative RMSE and the MSD decomposition of $\theta_s$ between GLEAM and ORCHIDEE. Low relative RMSE ($< 0.3$) covers most regions except for the North China Plain ($> 0.5$), where the MSD is dominated by the squared bias values (SB, Fig. 5(b)). This is clearly shown in the relative difference (Fig B7) between GLEAM and ORCHIDEE where simulated $\theta_s$ is approximately 30% lower than in GLEAM. Southern China has lower relative RMSE ($< 0.2$), and MSD is dominated by SB as well. Different from the North China Plain, SB in southern China may be due to the mismatch of land cover and soil parameterization between ORCHIDEE and GLEAM. For instance, the saturated SM in South China is 0.36 m$^3$.m$^{-3}$ while the maximum SM in GLEAM is 0.45 m$^3$.m$^{-3}$. A high contribution of LCS to MSD is found in Qinghai-Tibetan Plateau, the upper part of the YZRB and the YLRB, suggesting a mismatch of the phase of SM variability. The MSD is dominated by SDSD in northwestern China ($P < 200$ mm.yr$^{-1}$) suggesting different magnitudes of SM fluctuations. Nevertheless, the relative RMSE in Qinghai-Tibetan Plateau and northwestern China is as low as in southern China ($< 20\%$). Overall, ORCHIDEE is able to give a reasonable estimation of $\theta_s$ in regions where irrigation is not widespread.

Figure 6(a)-(e) shows NLD of ORCHIDEE and GLEAM $\theta_s$ computed based on the $k$-lag autocorrelation coefficient $R_k$. High NLD implies that $\theta_s$ has a longer memory in response to rainfall inputs. However, the spatial distribution of NLD depends not only on rainfall frequency and intensity but also on evapotranspiration and runoff losses after SM recharge by rainfall. The NLD patterns of GLEAM and ORCHIDEE $\theta_s$ are similar, which is encouraging in terms of how ORCHIDEE simulates the processes controlling the decrease of SM after each rainfall. Both southern and southeastern China have higher NLD, like in GLEAM. Lower NLD ($\approx 20$ days) prevail around 30°N in eastern China, whilst the North China Plain has NLD values of 40 days. The main difference of NLD between GLEAM and ORCHIDEE is in Inner Mongolia and over the Loess Plateau, where the ORCHIDEE NLD has values of 20 days, against 40 days in GLEAM. $R_k$ of spatially averaged $\theta_s$ in three regions is shown in Figure 6(f)-(h). Overall, $R_k$ of ORCHIDEE $\theta_s$ is consistent with that of GLEAM. The GLEAM $R_k$ is close to the ORCHIDEE $R_k$ in the YZRB with difference less than 6 days. In the YLRB, GLEAM $R_k$ is larger than ORCHIDEE $R_k$, suggesting that modeled $\theta_s$ has a faster response to rainfall input. Such bias can be explained by higher simulated evapotranspiration in the YLRB compared to GLEAM (Fig. B8), suggesting that the decline of ORCHIDEE $\theta_s$ is faster after rainfall events than in GLEAM and lead to a lower $R_k$.

The trend of ORCHIDEE $\theta_s$ (Fig. B9) is less significant than that of GLEAM $\theta_s$ (Fig. B9). In northwestern China, increasing $\theta_s$ is found in simulations ($< 0.2 \times 10^{-3}$m$^3$.m$^{-3}$.yr$^{-1}$) and GLEAM ($0.2$-$0.4 \times 10^{-3}$m$^3$.m$^{-3}$.yr$^{-1}$). The trend may be due to increasing $P$ (Fig. B10). GLEAM $\theta_s$ decreased dramatically in eastern China ([103-122°E]×[20-35°N]) while the trends of ORCHIDEE $\theta_s$ are not homogeneous in this region. In addition, all forcing datasets show an increasing $P$ in the North China Plain, which leads to slight increase of simulated $\theta_s$. But GLEAM shows decreasing $\theta_s$ in most area of the North China Plain. The mismatch of $\theta_s$ and $P$ trends suggests that the change of precipitation amount is not the only driver of the trend of SM.

## 4.2 Comparison of the four forcing datasets

To find the most realistic forcing dataset for SM performance given the ORCHIDEE model, several metrics were calculated and shown in Figure 7. Radar charts show the correlation coefficients ($r$) and RMSE of simulated SM in comparison to different datasets. Histograms show MSD and its three components. The median of specific metrics is listed in Table 3. GSWP3 has the best performance in estimating the magnitude of SM (lowest MSD) while WFDEI shows the best score in simulating SM variation (highest $r$). PGF provides as good estimation as GSWP3 in the YZRB, but performs more poorly in capturing SM variation in the YLRB, which is also reflected from the components of MSD. The largest MSD is found in CRU-NCEP in most of comparisons, which is mainly contributed by SB. The SDSD and LCS of CRU-NCEP are also larger than others but the differences are not as significant as SB. In addition, we performed the comparison over the full period (1981-2009). Corresponding metrics are shown in Table A1. The values vary slightly, but they do not change our conclusions.

Thus we conclude that both GSWP3 and WFDEI are suitable to simulate SM dynamics in China with ORCHIDEE. The best choice can be made based on the main focus of specific research. For estimating magnitude of SM, GSWP3 is preferable; for investigating SM variation, WFDEI is the best choice. Note that this study only provides the evaluation of SM, but other hydrological components should be compared with observations to confirm the superiority of GSWP3 and WFDEI.

## 4.3 Source of SM difference among simulations

By investigating the $D$ of meteorological variables and simulated SM among the four simulations ($D_x$ for variable $x$; Eq. 6), two questions are addressed: (1) How is $D$ of simulated SM and forcing variables spatially distributed? (2) Can spatial patterns of $D$ of SM be explained by that of meteorological variables? Note that the relative value of $D$, $D$ over the magnitude of specific variable in each grid cell, is not suitable for the analysis (detailed explanation is in Supplement C).

Figure B11 shows maps of $D$ of $\theta_t$ and meteorological variables. As the unit of $D$ depends on specific variables, it can only be used to compare spatial distributions, not values. High $D_{\theta_t}$ is found in southwest of China ([92-104°E]×[28-35°N]). However, similar patterns do not exist in the $D_P$ (Fig. B11(b)), suggesting that the difference of simulated $\theta_t$ is not caused by the difference of precipitation amount of forcing data. Similarly, in southwestern China, no high $D$ is found in meteorological variables except for number of precipitation days ($N_p$) and air humidity ($Q_a$), although the patterns of $D_{N_p}$ and $D_{Q_a}$ overlap with $D_{\theta_t}$ but extend to zones with low $D_{\theta_t}$ as well (Fig. B11(c) and (g)).

To look for clearer links between input and SM $D$, we decompose $N_p$ and $\theta_t$ by scales of $P$ and soil layers respectively (Sect. 3.4). The $r$ of $D$ between simulated SM and meteorological variables are shown in Figure 8. $D_{Q_a}$, $D_{N_p}$, and $D_{N_p^2}$ are highly correlated with $D_{\theta_t}$, implying that the difference of simulated $\theta_t$ can be explained by the differences of $Q_a$ and $N_p$ among the four forcing datasets. The $r$ between $D_{\theta_t}$ and $D_P$ is less than 0.3. All in all, the results suggest that the uncertainty of precipitation frequency and intensity is more important than that of precipitation amount in influencing SM differences among the simulations.

## 5    Discussion

### 5.1    Performance of the model to simulate SM

Due to the spatio-temporal complexity of SM and its vertical profile, four datasets were selected to drive the simulations and modeled SM at different depths was validated against multiple datasets. The results showed that ORCHIDEE SM coincides well with CCI (median $r = 0.48$; median RMSE $= 0.06$) and GLEAM SM (median $r = 0.55$; median RMSE $= 0.07$) in comparison to other model studies (Lai et al., 2016).

Higher $r$ were systematically found in southern China, the Loess Plateau, and the North China Plain; lower $r$ were found in northwestern China, western Tibetan Plateau, eastern Sichuan basin and downstream of the YZRB. SM is underestimated significantly in the Loess Plateau and the North China Plain with modeled values being 20% and 30% less than in CCI and GLEAM, respectively (Fig. B7 and B3). It is not only due to model parameterization but also due to irrigation activities in those agricultural regions (Fig. 1), which are not considered in the simulations.

Because the in-situ SM measurements were collected only for croplands and grasslands (Piao et al., 2009; Robock et al., 2000), implying potential disturbances from human activities, $r$ was low in the comparison to ISMN and PKU datasets (median $r = 0.37$, Fig. 2). For instance, drought occurred in northern China during 1987-1988 (Yang et al., 2012), which is reflected in the variation of measured SM at Xifeng and Xinxian (Fig. 3(a)-(b)). ORCHIDEE successfully reproduced the drought induced SM decline at the two stations. But SM measured at Xinxian was maintained at a high level. A possible explanation is that the soil at Xinxian was irrigated. Consequently SM at Xinxian did not vary with precipitation leading to a low $r$ ($< 0.23$). Another possible reason leading to the mismatch between simulations and in-situ measurements is scale effects. Local measurements can be an ideal choice for model validation only if the atmospheric forcing was provided at the same scale due to the spatial variability of precipitation and of landscape. Otherwise remote sensing products derived from multiple observations averaged or aggregated at daily time step are probably more comparable to model simulations obtained with meteorological reanalysis than local in-situ measurements.

In the comparison to CCI and GLEAM SM, low $r$ did not occur in the northern China, such as the Loess Plateau and the North China Plain, but was found in the climatic transition zone between southern and northern China (Fig. 4, B5, and 5). Irrigation may strongly influence SM dynamics in northern China, and in turn reduce $r$. However, such effect to $r$ is not significant because of the large seasonality of SM in this region. Instead of $r$, the impacts of irrigation are mainly reflected from the RMSE and relative difference (Fig. B2, B3, B6, and B7). Thus for a region with both irrigation and strong seasonality, bias and RMSE are recommended to trace the footprint of irrigation rather than correlation coefficients. In the climatic transition zone (e.g., Sichuan Basin, mid- and down-stream of the YZRB), climatic seasonality is not as large as the northern China. Meanwhile irrigation is still needed for agriculture, which consequently results in low $r$ between simulated and observed SM.

From the results we conclude that ORCHIDEE provides a satisfactory simulation of SM dynamics in China, except in areas subject to irrigation. This calls for inclusion of irrigation and realistic crop phenology (Wang et al., 2017) as a priority for future application of this model for SM and river discharge dynamics.

## 5.2 Linkage of discrepancies between meteorological factors and SM through ORCHIDEE

In Section 4.3, we showed that the spatial differences of simulated SM among the four forcing datasets were highly correlated with forcing differences in $N_p$ and $Q_a$. This suggests that the uncertainty of precipitation frequency is more critical than that of precipitation amount in determining variation of SM patterns, as pointed out by other studies, especially in arid and semi-arid regions (Baudena and Provenzale, 2008; Cissé et al., 2016; Piao et al., 2009). To precise the result, we studied the correlation coefficients between the spatial averaged $D$ of SM in different soil layers and of $N_p$ categorized by classes of precipitation intensity (Fig. 8). The result showed that differences in small rainfall events $N_p^i$ with $1 < P < 5$ mm.d$^{-1}$ are more important than other precipitation classes in explaining SM differences due to atmospheric forcing datasets.

Differences in $Q_a$ were also shown to explain a large fraction of the simulated SM differences across different forcings. $Q_a$ determines vapor pressure deficit, which in turn controls transpiration (Farquhar and Sharkey, 1982) and evaporation (Monteith, 1965), suggesting a strong control by atmospheric dryness of the differences in SM found among the four forcing datasets. Both of $Q_a$ and $N_p$ have positive impacts on SM, which enhances the correlation in Figure 8.

Estimating impacts of meteorological factors on SM dynamics is difficult. First of all, the importance of a meteorological variable on SM may vary with climate regimes. For instance, the importance of precipitation and radiation on SM changes from water to energy limited regions. Secondly, impacts of meteorological variables can be nonlinear through interactions with local ecosystem (Seneviratne et al., 2010), suggesting that even with same meteorological variable the simulated SM can be totally different (e.g., with different soil texture or land cover types). Moreover, SM can be strongly coupled with atmosphere (Koster, 2004; Taylor et al., 2012), implying that meteorological factors can be influenced by SM as well (such as cloudiness, precipitation, air humidity, etc), which is not included in this study. However, the logic of our importance analysis is simple. If the model inputs (forcing data) were the same, the outputs (SM) should be the same. In other words, the differences of outputs can only be caused by the difference of inputs in our simulation results. It does not matter whether the quality of atmospheric forcing is good. On the contrary, the more differences exists among these forcing datasets, the better our analysis is. To keep the analysis simple, we did not investigate temporal correlations in each pixel but focused on spatial patterns of $D$ at continental scale. Therefore, our results provided a general estimation of the importance of meteorological variable uncertainties to SM simulation through ORCHIDEE.

Indeed this approach is not able to demonstrate explicit links between meteorological variables and SM. We underlined the impacts of $N_p$ and $Q_a$ uncertainties, but it does not mean that other factors are unimportant. For instance, assuming that a variable can strongly influence simulated SM, if there was no much difference of the variable among forcing datasets, its importance cannot be detected in this work. Moreover, only one model was used in this study. Although ORCHIDEE performed very well in SM simulation, the lack of unknown mechanisms may weaken the linkage between SM and specific atmospheric variables. In one word, our analysis only focused on the inputs and outputs of the model and tried to diagnose the relationship between their differences.

## 6    Conclusions

Simulations in China were performed in ORCHIDEE-MICT driven by different forcing datasets: GSWP3, PGF, CRU-NCEP, and WFDEI. Simulated soil moisture was compared to several datasets to evaluate the ability of ORCHIDEE-MICT in reproducing soil moisture dynamics in China. Results showed that ORCHIDEE soil moisture coincided well with other datasets in wet areas and in non-irrigated areas. It suggested that the ORCHIDEE-MICT was suitable for further hydrological studies in China. However, the abnormal variation of observed SM in North China Plain implied potential impacts of irrigation, which was recommended to be considered in further simulations. Moreover, results showed that bias was mainly from model parameterization and atmospheric forcing. Thus parameterizations in ORCHIDEE-MICT should be calibrated, and atmospheric forcing should be carefully selected to meet the situation of China.

Several criteria were chosen and compared among the four simulations in China, YZRB, and YLRB. Results showed that GSWP3 and WFDEI, which had the best performances in correlation coefficients and RMSE respectively, were ideal choices for hydrological study in China. However, higher MSD in the Yellow River basin reflected the complicated climate condition in northern China, which might be significantly influenced by human activities as well. Finally, we used the differences of simulated soil moisture and meteorological variables to simply investigate the linkage between them. Results showed that the differences of simulated soil moisture were mainly explained by the differences of air humidity and precipitation frequency among the four atmospheric forcing. However, this coarse analysis cannot give explicit explanations about related mechanisms. Further study is needed to discover the interactions between soil water and climate through tracing the surface hydrological cycles and energy balances.

*Code and data availability.*    The SVN version of ORCHIDEE-MICT used in this study is 3952, which is available at https://forge.ipsl.jussieu. fr/orchidee/wiki/DevelopmentActivities/ORCHIDEE-MICT-IMBALANCE-P. The ORCHIDEE code and scripts of analysis are available by contacting the correspond author. The GSWP3, PGF, WFDEI, ISMN, GLEAM and ESA CCI datasets are freely available online; for the CRU-NCEP and PKU datasets, please contact the corresponding author and Prof. Shilong Piao (slpiao@pku.edu.cn) respectively.

Members of the China-Trend-Stream project are (alphabetically): Philippe Ciais[1], Patrice Dumas[2], Xiaoming Feng[3], Matthieu Guimberteau[1,4], Laurent Li[5], Catherine Ottlé[1], Shushi Peng[6], Shilong Piao[6], Jan Polcher[5], Pengfei Shi[7], Shuai Wang[3], Xuhui Wang[1,5,6], Yi Xi[6], Hui Yang[6], Tao Yang[7], Zun Yin[1], Xuanze Zhang[6], Feng Zhou[6], and Xudong Zhou[7].

[1]Laboratoire des Sciences du Climat et de l'Environnement, CNRS-CEA-UVSQ, Gif-sur-Yvette 91191, France

[2]Centre de Coopération Internationale en Recherche Agronomique pour le Développement, Avenue Agropolis, 34398 Montpellier Cedex 5, France

[3]State Key Laboratory of Urban and Regional Ecology, Research Center for Eco-Environmental Sciences, Chinese Academy of Sciences, Beijing 100085, China

[4]UMR 7619 METIS, Sorbonne Universités, UPMC, CNRS, EPHE, 4 place Jussieu, Paris 75005, France

[5]Laboratoire de Météorologie Dynamique, UPMC/CNRS, IPSL, Paris 75005, France

[6]Sino-French Institute for Earth System Science, College of Urban and Environmental Sciences, Peking University, Beijing 100871, China

[7]State Key Laboratory of Hydrology-Water Resources and Hydraulic Engineering, Center for Global Change and Water Cycle, Hohai University, Nanjing 210098, China

5    *Author contributions.*  P. Ciais, C. Ottlé and Z. Yin designed research. Z. Yin performed research, analyzed data and wrote the draft; all authors contributed to interpreting results, discussing findings and improving the manuscript.

*Competing interests.*  The authors declare that they have no conflict of interest.

*Acknowledgements.*  This study was supported by the National Natural Science Foundation of China (grant number 41561134016) and by the CHINA-TREND-STREAM French national project (ANR Grant No. ANR-15-CE01-00L1-0L). M. Guimberteau and P. Ciais acknowledge
10   support from the European Research Council Synergy grant ERC-2013-SyG-610028 IMBALANCE-P. H. Kim was supported by Japan Society for the Promotion of Science KAKENHI (16H06291). We thank Brecht Martens and Suxia Liu for helpful discussion about GLEAM and ISMN datasets. We gratefully acknowledge two anonymous referees and the editor for their helpful comments and efforts.

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

## Tables

**Table 1.** General information of the climate forcing datasets. 'Reanalysis' and 'Observations' are corresponding datasets used in producing the atmospheric forcing. Detailed description can be found in Sect. 2.2.

| Dataset | Resolution | | Duration | Reanalysis | Observations |
|---|---|---|---|---|---|
| | Spatial | Temporal | | | |
| GSWP3 | 0.5° | 3-hourly | 1901–2010 | 20CR | GPCC, CRU TS, SRB |
| PGF | 1° | 3-hourly | 1901–2012 | NCEP-NCAR | CRU TS, GPCP, TRMM, SRB |
| CRU-NCEP | 0.5° | 6-hourly | 1901–2015 | NCEP | CRU TS |
| WFDEI | 0.5° | 3-hourly | 1979–2009 | ERA-Interim | CRU TS, GPCC |

**Table 2.** Summary of the SM datasets for validation. 'M+RS+RA' indicates that the dataset is a model output driven by both remote sensing and reanalysis data. More details can be found in Sect. 2.3.

| Dataset | Type | Unit | Resolution | Duration | Contents | Corresponding |
|---|---|---|---|---|---|---|
| | | | | Analysis period | Analysis depth | ORCHIDEE soil layer |
| ISMN | in-situ | $m^3.m^{-3}$ | station, 10-day | 1981-1999 | 11 layers; 0-100 cm | 1-9 layers (0-75 cm) |
| | | | | 1984-1999 | 0-50 cm | |
| PKU | in-situ | % of porosity | station, 10-day | 1991-2007 | 7 layers; 0-100 cm | 1-8 layers (0-37 cm) |
| | | | | 1992-2006 | 0-30 cm | |
| ESA CCI | RS | $m^3.m^{-3}$ | 0.25°, daily | 1979-2010 | Top layer, depth ≈ 0.5-2 cm | 1-4 layers (0-2 cm) |
| | | | | 2007-2009 | | |
| GLEAM surface | M+RS +RA | $m^3.m^{-3}$ | 0.25°, daily | 1980-2014 | 0-10 cm | 1-6 layers (0-9 cm) |
| | | | | 1981-2009 | | |
| GLEAM root-zone | M+RS +RA | $m^3.m^{-3}$ | 0.25°, daily | 1980-2014 | Mixture of bare soil (1-10 cm), low vegetation (0-100 cm) and high vegetation (0-250 cm) | all layers (0-200 cm) |
| | | | | 1981-2009 | | |

**Table 3.** Median of metrics in specific comparisons. The subscripts of correlation coefficients indicate the quantile of stations (samples) with significant correlation ($p$-value $< 0.05$).

| Dataset | Simulations | Correlation | | | RMSE ($m^3.m^{-3}$) | | |
|---|---|---|---|---|---|---|---|
| ISMN | GSWP3 | $0.52_{0.85}$ | | | 0.07 | | |
| | PGF | $0.46_{0.90}$ | | | 0.07 | | |
| | CRU-NCEP | $0.36_{0.95}$ | | | 0.10 | | |
| | WFDEI | $0.55_{0.95}$ | | | 0.08 | | |
| PKU | GSWP3 | $0.38_{0.91}$ | | | NA | | |
| | PGF | $0.31_{0.85}$ | | | NA | | |
| | CRU-NCEP | $0.31_{0.86}$ | | | NA | | |
| | WFDEI | $0.45_{0.93}$ | | | NA | | |
| | | China | Yangtze | Yellow | China | Yangtze | Yellow |
| ESA CCI | GSWP3 | $0.47_{0.93}$ | $0.42_{0.94}$ | $0.58_{0.99}$ | 0.06 | 0.06 | 0.06 |
| | PGF | $0.26_{0.83}$ | $0.32_{0.91}$ | $0.28_{0.96}$ | 0.06 | 0.07 | 0.07 |
| | CRU-NCEP | $0.51_{0.94}$ | $0.50_{0.94}$ | $0.54_{0.99}$ | 0.06 | 0.07 | 0.07 |
| | WFDEI | $0.61_{0.97}$ | $0.60_{0.96}$ | $0.68_{1}$ | 0.05 | 0.05 | 0.06 |
| GLEAM surface SM | GSWP3 | $0.54_{1}$ | $0.60_{1}$ | $0.52_{1}$ | 0.07 | 0.07 | 0.10 |
| | PGF | $0.42_{1}$ | $0.51_{1}$ | $0.35_{1}$ | 0.08 | 0.08 | 0.10 |
| | CRU-NCEP | $0.49_{0.99}$ | $0.61_{1}$ | $0.49_{1}$ | 0.10 | 0.10 | 0.12 |
| | WFDEI | $0.68_{0.99}$ | $0.77_{1}$ | $0.63_{1}$ | 0.08 | 0.09 | 0.10 |
| GLEAM root-zone SM | GSWP3 | $0.60_{0.98}$ | $0.67_{0.99}$ | $0.60_{0.99}$ | 0.05 | 0.04 | 0.08 |
| | PGF | $0.57_{0.98}$ | $0.69_{1}$ | $0.57_{0.99}$ | 0.06 | 0.04 | 0.09 |
| | CRU-NCEP | $0.40_{0.96}$ | $0.48_{0.97}$ | $0.37_{0.97}$ | 0.08 | 0.08 | 0.11 |
| | WFDEI | $0.63_{0.98}$ | $0.74_{1}$ | $0.59_{1}$ | 0.06 | 0.04 | 0.10 |

**Figures**

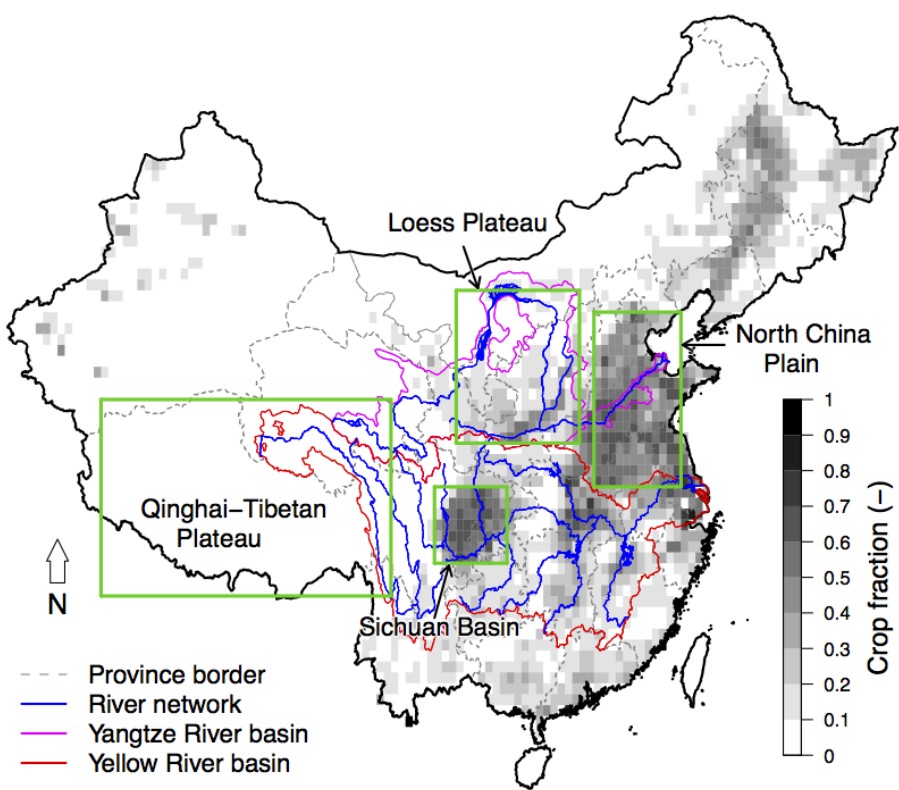

**Figure 1.** Map of China. The grey background is cropland fraction. Green rectangulars show four important regions mentioned in this paper.

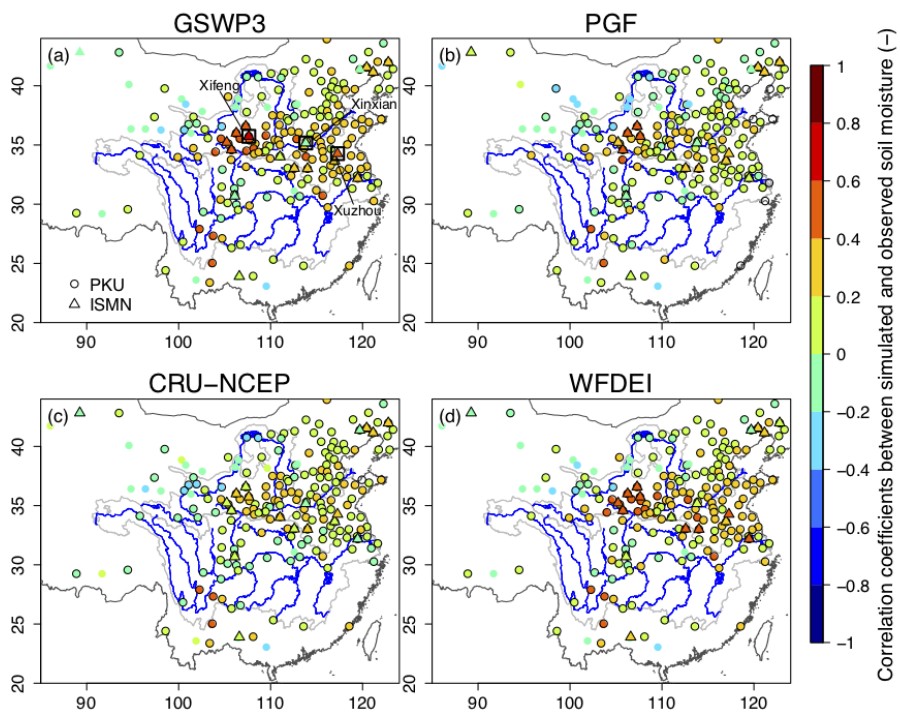

**Figure 2.** Pearson correlation coefficients of modeled and measured SM at each gauging station from ISMN (triangles) and PKU (circles). Symbols with dark border indicate significant correlations ($p < 0.05$). The locations of three ISMN stations shown in Figure 3 are marked by black squares in panel (a).

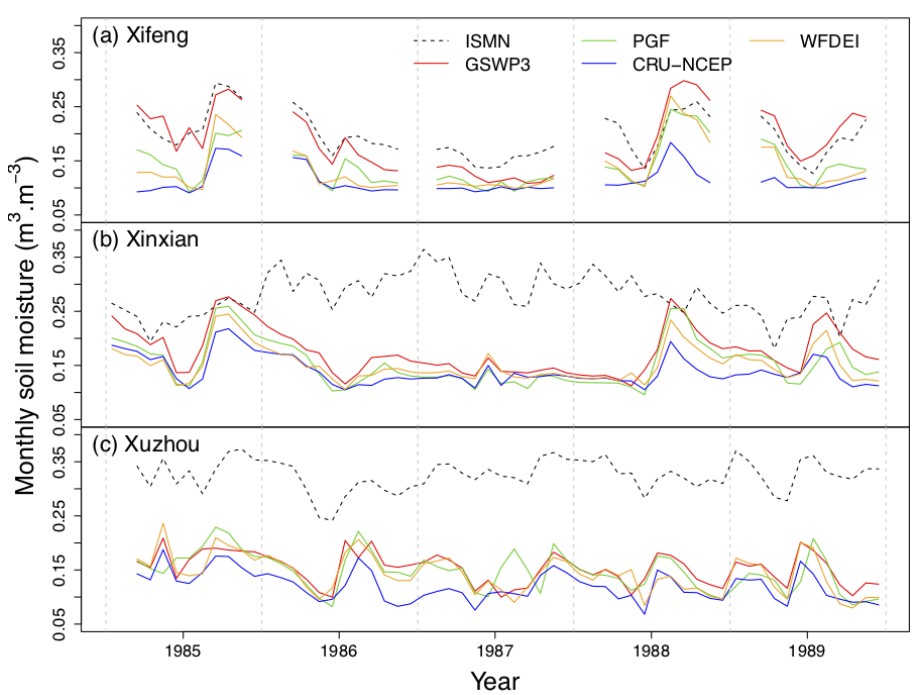

**Figure 3.** Time series of 10-day SM from ORCHIDEE and ISMN at three stations. The station locations are shown in Fig. 2(a). The mean annual precipitation at Xifeng, Xinxian, and Xuzhou (according to GSWP3) are 556, 580, and 847 mm.yr$^{-1}$, respectively. Dark dashed lines indicate ISMN SM. Red, green, blue, and orange lines indicate simulated SM based on GSWP3, PGF, CRU−NCEP, and WFDEI, respectively.

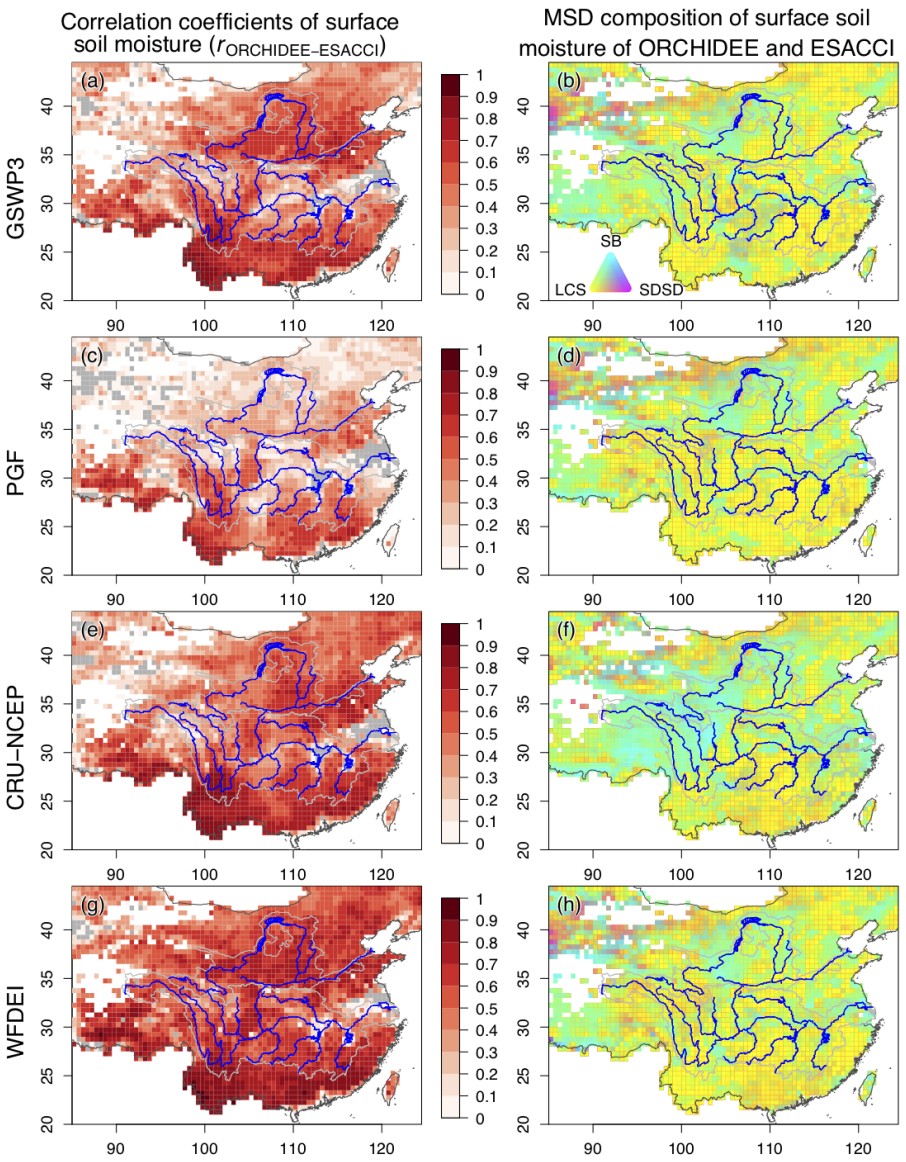

**Figure 4.** Left panel: Correlation coefficients of the ESA CCI SM and the corresponding ORCHIDEE SM. Gray pixels indicate non and negative correlation. Right panel: decomposition of the MSD between the daily ESA CCI SM and the corresponding ORCHIDEE SM (Eq. 1). Cyan, magenta, and yellow indicate the fractions of SB, SDSD, and LCS respectively.

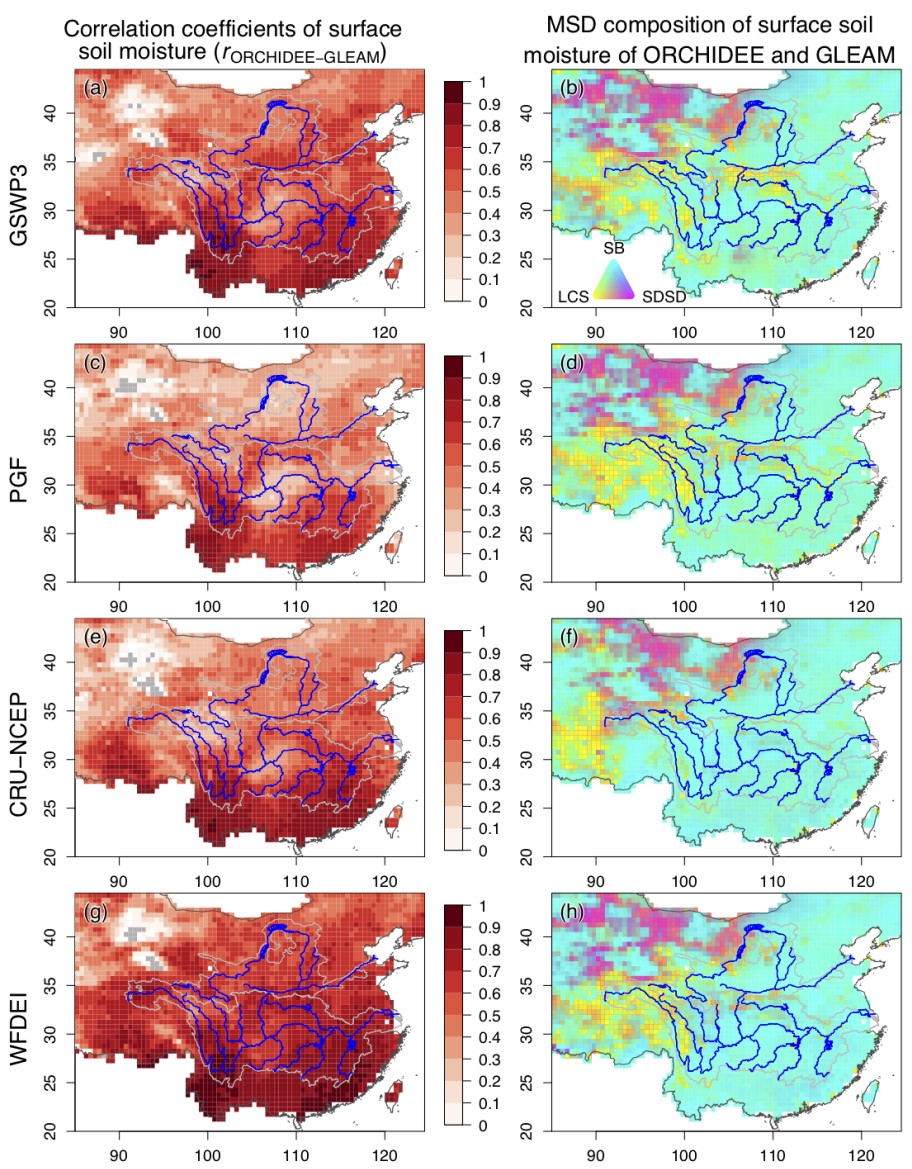

**Figure 5.** Left panel: Pearson correlation coefficients of the GLEAM surface SM and the corresponding ORCHIDEE SM. Gray indicates non and negative correlation. Right panel: decomposition of the MSD between the daily GLEAM surface SM and the corresponding ORCHIDEE SM (Eq. 1). Cyan, magenta, and yellow indicate the fractions of SB, SDSD, and LCS respectively.

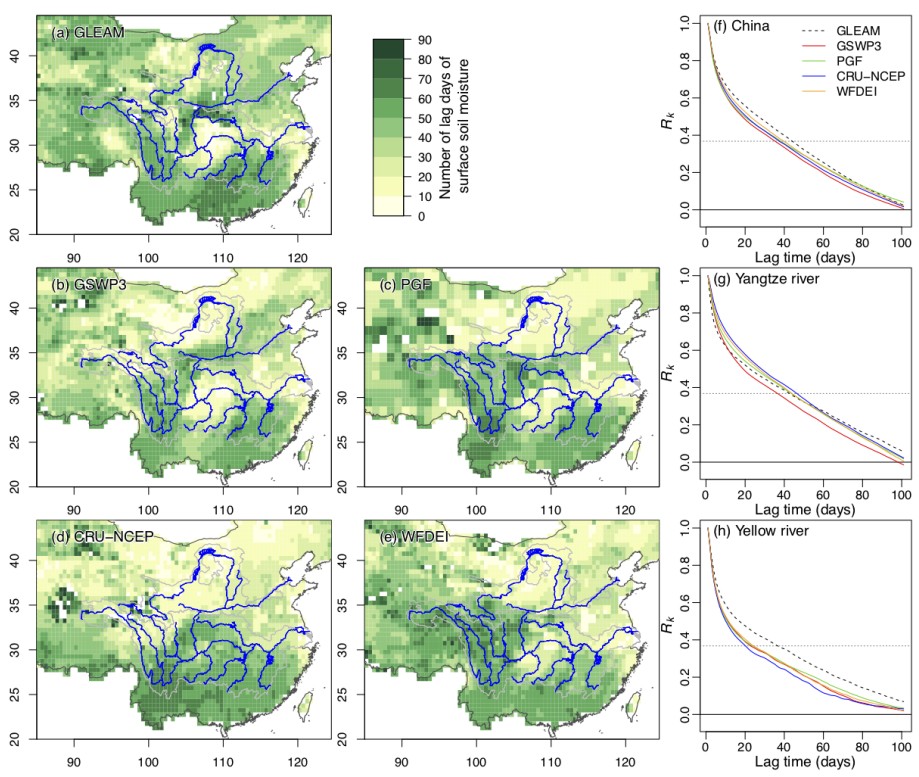

**Figure 6.** (a): Number of lag days (NLD) of GLEAM surface SM. (b)-(e): Difference of NLD between GLEAM and ORCHIDEE surface SM. (f)-(h): Autocorrelation coefficient $R_k$ of spatial averaged surface SM as a function of NLD. The dashed line ($y = 1/e$) is the threshold of significant correlation.

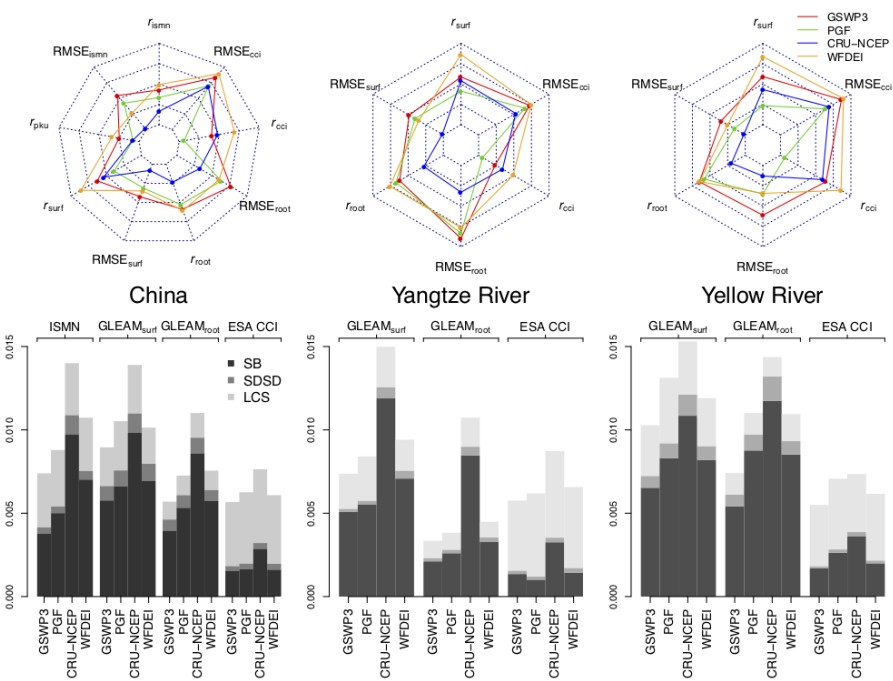

**Figure 7.** Evaluation of the forcing datasets for simulating SM dynamics in China, YZRB, and YLRB. Top panel: Radar charts of criteria of the four forcing datasets. Center implies bad criteria. Red, green, blue, and orange lines indicate GSWP3, PGF, CRU-NCEP, and WFDEI, respectively. 'surf' and 'root' indicates surface and root-zone SM of GLEAM 3.0A. Bottom panel: Composition of MSD from each comparison. $x$-axis indicates the drivers of specific simulations; top labels indicate the data set used in the specific comparison.

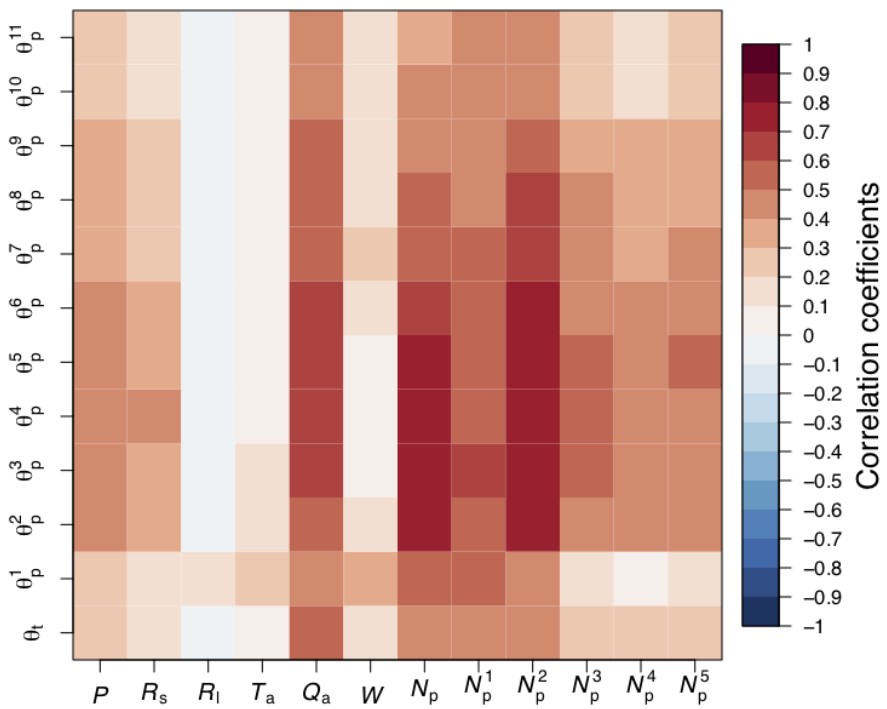

**Figure 8.** Matrix of correlation coefficients between the $D$ of meteorological variables and the $D$ of simulated SM. $D$ is the averaged MSD defined by Eq. 6. $\theta_t$ indicates total SM. $\theta_p^i$ indicates SM in $i$th layer. $P$ indicates annual precipitation. $R_s$ and $R_l$ indicate short and long wave incoming radiation, respectively. $T_a$ indicates air temperature. $Q_a$ indicates air humidity. $W$ indicates wind speed. $N_p$ indicates the number of days with precipitation no less than 0.01 mm.d$^{-1}$. $N_{p^i}$ indicates the number of days with a specific precipitation range (Sect. 3.4).