# Peer review of "Evaluation of ORCHIDEE-MICT simulated soil moisture over China and impacts of different atmospheric forcing data"

_Hydrology and Earth System Sciences, 2017_

## Referee Comment (RC1) · Anonymous Referee #1 · 28 Feb 2018

General Comments

The paper systematically explores the impact of four different atmospheric forcing datasets on soil moisture simulations across China, computed by the ORCHIDEE-MICT land surface model.

Studies such as this one are important to understand the considerable differences caused by different forcing datasets prior to interpreting model output of a specific forcing / land surface model combination. The paper is well written and presented in good English. The graphics are very well designed.

The choice of the four different datasets seems reasonable, namely the GSWP3, PGF,

[Figure]

CRU-NCEP and WATCH dataset, and the choice of these datasets seems to show that the authors want to base the validation statistics on multi-year / decadal analysis.

Some criticism:

The comparison against CCI soil moisture only for 2007 until 2009 is a odd choice. CCI is a unique soil moisture dataset in being based on observations and covering a long time period. This makes it different to other available long-term soil moisture datasets based on model output and other observation based datasets which are usually much shorter. Therefore it should be taken as what it was designed for without cherry-picking the best period. Also, these long time periods will be likely much more interesting for most readers as a limited amount of specific years.

Also, the comparison is not too meaningful if the other datasets / experiments are not compared for the same time period.

Please make the choice of GLEAM clearer. It uses a lot of observations but it essentially is also model output. So you are comparing your model output to another model (which uses a different precipitation forcing?) Possibly give a little more literature on other soil moisture datasets, why specifically GLEAM, e.g. long time period?

Describe why you specifically chose those four forcing datasets. Are they being frequently updated? Also usable for global studies?, etc.

The motivation of carrying out the study specifically over China is in my view lacking a little. Also, are there no locally optimised forcing datasets available? Why run a land surface model specifically over China using global input data? Again, just make the motivation of the study a bit clearer. Why was this specific model used for the experiments, does it have any advantages specifically for China (this is actually mentioned in the model section but might be also helpful in the introduction with a little more detail)?

Concerning the validation as a whole, the model outputs for the four experiments are compared to, in addition to in-situ measurements, GLEAM and CCI soil moisture. How-
ever, these datasets (CCI and GLEAM) should also be compared to the in-situ measurements since the mere comparison does not result in any helpful answer on which of these datasets performs any better when compared to the actual ground measurements. Both GLEAM and the CCI dataset will likely have their own problems with accurately simulating soil moisture within certain areas. At the current state of the study they are used as a kind of additional ground-truth, which they most certainly are not (and in fact, as correctly noted, GLEAM shares some of the same input data with the forcing data used for the experiments).

Given these shortcomings I advise for a major revision.

Specific Comments

1 Introduction

P2L21-32: Possibly add a sentence on soil moisture (products or raw data) data assimilation in the introduction, since the advantage / disadvantages of satellite based soil moisture products and land surface models are discussed. Data assimilation exactly tries to combine the strengths of these different types of data, such as in the GLEAM dataset.

Dataset description

2.1. Atmospheric forcing

P3L22-29: GSWP3 is very coarse, but downsampled. Could this be especially problematic in areas within China with complex terrain?

P4L2-8: Is PGF still being updated? Maybe add this information to the other datasets too, or to the motivation of choosing these specific datasets.

P4L18-22: WFDEI, why only available until 2009? Both corrected with GPCC v5 and v6?

2.2 Soil moisture datasets

P5L20: GLEAM has $0.5°$ resolution? I thought 0.25. It's $0.25°$ in Table 2, please recheck.

P5L28 GLEAM assimilates GLDAS? I'm not so sure about this. I think it's somehow used for the background error estimation within the assimilation scheme, but please check this.

3) Land surface model, ..

P6L26: 13 PFTs are grouped, did not understand. Only three land cover classes?

P7L10. Why aggregate results to 1 degree? This likely deteriorates the impact of "high resolution" forcing datasets, such as WFDEI. Rather upsample coarser data by simply multiplying grid cells?

P7L15: "distributed to the first half of the forcing time step.." why the first half.

3.3) Model-data comparison methodology and metrics

Comparison protocol

Metrics

P8L1-3: Which time shift was used? Between UTC and local time (several time zones) between model and in-situ measurements. Not vital but good to know.

P8L24: What is the exact motivation for the lag analysis? It does not seem to give any added value. How do you know one or the other are better in temporal terms? You are comparing two models.

3.4) Correlation of uncertainties between SM and meteorological factors

P9L12: Monthly values of other variables also considered . . . How?

4 Results

4.1. Spatial patterns of precipitation and simulated soil moisture

P9L23: These two rivers are the main ones? How much of China do these two watersheds cover? Maybe obvious for some but more background on the study region could be valuable (here and / or in the introduction).

This seasonality is computed across the boundaries depicted in Figure 1? A little more geographically distributed information would be helpful.

P10L3: The soil moisture patterns do not necessarily match the annual mean precipitation patterns, maybe mention something about obvious monthly differences, or stronger evaporation using a specific forcing dataset? Soil moisture is not just the result of precipitation but also the other input data and model internal mechanics. No in-depth analysis is needed but some additional maps or statistics for the most important other water balance variables, e.g. evapotranspiration, or at least some sentences on the issue would be helpful. The GLEAM model you are comparing to is actually primarily developed for evapotranspiration.

4.2. Soil moisture evaluation against multiple datasets, etc..

This part could benefit from some restructuring:

"Comparison with ISMN and PKU in-situ data" seems to be a summary of the model performance for all four forcing datasets when compared to in-situ measurements. It should be noted that these are the average statistics for all carried out experiments. Instead of the next section being "Comparison with GLEM ..." I as a reader would expect a more detailed analysis to follow (or the other way round), which now seems to be in section 4.3 and 4.4. Thus I would recommend to first do the in-depth comparison to in-situ measurements, followed by a comparison to other datasets thereafter. As stated at the beginning, I strongly believe that GLEAM and the CCI dataset should be validated against the in-situ measurements if you want to quantify which model actually performs better in which geographical area.

The main finding that WFDEI performs best among forcing data is not so surprising

when compared to some other studies. Again, more emphasis should be put on why this study is important specifically for China. Maybe compare the outcome of the study to other studies.

Figures:

Table 2: Correlations are stated as being significant. Was the autocorrelation of the datasets taken into account? Also valid for the correlation at the individual stations.

Figure 10: Should include description of variable names. Use same variable names in Figures 9 and 10.

---

## Referee Comment (RC2) · Anonymous Referee #2 · 23 Apr 2018

**OVERVIEW**

The manuscript analyses the capability of different atmospheric forcing datasets in reproducing soil moisture over China through ORCHIDEE-MICT model. Specifically, four atmospheric forcing datasets (GSWP3, PGF, CRU-NCEP, WFDEI)) are considered and the corresponding modelled soil moisture data are compared with satellite soil moisture datasets (GLEAM and ESA-CCI) to assess their accuracy. A sensitivity analysis for trying to understand the meteorological variables influencing soil moisture variability

between datasets is finally carried out.

**GENERAL COMMENTS**

The manuscript is well written and clear. The topic is interesting for the readership of HESS as the simulation of soil moisture at continental scale is important to understand the role of this important hydrological variable in governing the land-atmosphere interactions. Therefore, I believe the paper deserves to be published. However, some aspects should be improved, in my opinion, before the publication. I listed below the general comments (in order of appearance in the text) with also their importance.

1) MAJOR: The abstract contains some details that cannot be understood by reading the abstract only (it should be avoided). For instance, median R and RMSE are reported at page 1 - line 9 without mentioning with respect to which dataset they are computed. The reference to SB and LSC metrics is given but the reader is not able to understand what these metrics represent. Why are they used? Similarly for the discrepancies metric. I suggest mentioning in the abstract the results in general terms, without referring to metrics not know to the reader.

2) MINOR: Acronyms and symbols should be specified the first time they appear in the text, please check.

3) MAJOR: The selection of the reference datasets for soil moisture simulations might be questionable. Please try to fix the problem.

a. GLEAM contains several datasets included in the atmospheric forcing datasets. It

is not only ERA-Interim but also GPCC through MSWEP product. Therefore, I expect a large agreement between GLEAM and modelled soil moisture, but it does not mean the soil moisture simulations are accurate, they are simply consistent with GLEAM soil moisture (as I expected). The corresponding results should be clarified and put in perspective.

b. Even though it is a satellite-based dataset (therefore, its accuracy might be not good enough), the use of ESA CCI soil moisture dataset is in my opinion good. However, why only 3 years? I agree with authors that ESA CCI soil moisture product is more accurate after 2007, but for modelling assessment, I would prefer to see a long-term comparison (1980-2017). It is highly needed and to me much more appropriate than using GLEAM.

4) MODERATE: Too many figures, also by considering the Appendix, have been presented in the paper. I would prefer a lower number of more focused figures that would help the reader to understand clearly the main results the authors want to convey. Please try to reduce the length of the paper, mainly the results section.

5) MODERATE: The sensitivity analysis linking soil moisture and meteorological variables seems to me not robust enough for being published on HESS. I might be wrong, but also the authors acknowledge this problem. I suggest removing or, at least, strongly reducing.

**SPECIFIC COMMENTS**
Page 5, line 3: How is it assessed the quality of ISMN stations? Please clarify.

Page 5, line 17: I would not say "only" 203 stations.

Page 7, line 19: Soil depths are not different in the four datasets. If I am right, please

remove.

Page 10, line 16: "an traditional" should be "a traditional".

Page 10, line 20: Why the magnitude of soil moisture is systematically underestimated? Please try to find an explanation.

Page 12, lines 2-3: Again, why changes of precipitation regimes are not enough to predict changes in soil moisture? Please comment.

**RECCOMMENDATION**

On this basis, I found the topic of the paper relevant and interesting. Therefore, I suggest a moderate revision before the publication in Hydrology and Earth System Sciences.

---

## Author Comment (AC1) · 29 Jun 2018

**Reply to Referee #1 for "Evaluation of ORCHIDEE-MICT simulated soil moisture over China and impacts of different atmospheric forcing data" on HESSD**

Z. Yin on behalf of all co-authors

**1 General comments**

**1.1** **"The comparison against CCI soil moisture only for 2007 until 2009 is a odd choice. CCI is a unique soil moisture dataset in being based on observations and covering a long time period. This makes it different to other available long-term soil moisture datasets based on model output and other observation based datasets which are usually much shorter. Therefore it should be taken as what it was designed for without cherry-picking the best period. Also, these long time periods will be likely much more interesting for most readers as a limited amount of specific years. Also, the comparison is not too meaningful if the other datasets experiments are not compared for the same time period."**

A: The primary aim of the comparison using ESA CCI soil moisture (SM) is to assess our model outputs. The availability of ESA CCI SM varies a lot due to changes in sensors. Figure R1 shows the fraction of days with valid observations in different periods from Dorigo et al. (2015). It is clear that the fraction is extremely low in China (less than 0.2) until 2006. If we zoom in China and check CCI SM time series at some grid cells (Fig. R2), the availability varies not only temporally but also spatially. Once again, as our aim is to evaluate ORCHIDEE simulated soil moisture (spatial and temporal patterns), we have to select a period which presents the less gaps in the time series in order to be able to compare our evaluation metrics and their spatial variations. This explains why we choose the more recent period in the dataset, covering our simulations. In the manuscript, we modified as: "The data availability also varies along the period according to the number of instruments available and the increase of their temporal and spatial resolutions. In China, the fraction of days with available records (Figure 4 of Dorigo et al. (2015)) is lower than 20% from 1979 to 2006. More importantly, large spatial variation of gaps exists before 2006 (Fig. A1). ... To provide a reliable validation, we only use the CCI data between 2007-2009."

However, we agree that it is interesting for readers to see a long time period comparison. Thus we will provide the comparison between ESA CCI and ORCHIDEE SM from

1981 to 2009 in online supplementary, as Figure R3 and Table R1. Comparing to the same analysis based on period 2007-2009 (Fig. 7 and Table 3 in the main text), there is no significant change of spatial patterns. Although the values of $r$ and RMSE change slightly, they do not influence our conclusion that GSWP3 and WFDEI performance better then the other two. If both reviewer think that the long time period comparison is more important, we will thow the 1981-2009 comparison it in the revised manuscript instead.

**1.2 "Please make the choice of GLEAM clearer. It uses a lot of observations but it essentially is also model output. So you are comparing your model output to another model (which uses a different precipitation forcing?) Possibly give a little more literature on other soil moisture datasets, why specifically GLEAM, e.g. long time period?"**

A:True. GLEAM SM is a model output, but it is corrected by numerous satellite and in-situ measurements through data assimilation. The comparison to GLEAM SM is to provide an assessment of SM dynamics at longer time period. In the introduction, we will explain the aim of using different SM datasets, as: "The resulting SM is evaluated by different SM datasets including in-situ, remote sensing measurements and reanalysis. In-situ measurements including ISMN (International Soil Moisture Network; Dorigo et al. (2011)) and PKU (in-situ SM from Peking University; Piao et al. (2009); Xu (2014)) are used to evaluate temporal validation of ORCHIDEE SM. To evaluate spatio-temporal variations of simulated SM, the satellite based dataset ESA CCI SM (European Space Agency Climate Change Initiative Soil Moisture; Wagner et al. (2012)) is applied in the comparison. Note that both in-situ and satellite SM datasets represent the 'truth' to some extent. This implies that real-world soil moisture is influenced by processes that are not modeled such as irrigation and wetlands. Thus mismatches between measured and simulated SM may exist in some regions strongly affected by anthropogenic factors."
"Finally the GLEAM SM data (The Global Land Evaporation Amsterdam Model; Martens et al. (2017)) is compared to the simulated SM. Different from other SM datasets, GLEAM SM results from a land surface model constrained with a number of satellite and in-situ observations. This is not a direct observation but GLEAM was shown to reproduce reasonable long period SM dynamics at global scale (Martens et al., 2017), which is valuable to evaluate ORCHIDEE simulations for both surface and root zone moisture. Furthermore, GLEAM assimilates CCI data, so that evaluation of our model against root zone moisture from GLEAM is consistent with evaluation against surface moisture from CCI. Details of the SM datasets are shown in Sect. 2.3."

**1.3 "Describe why you specifically chose those four forcing datasets. Are they being frequently updated? Also usable for global studies?, etc."**

A: These four datasets are widely used in large scale hydrological studies, which are suitable for global simulation for next step. In the introduction, we added: "Four global atmospheric forcing datasets are chosen to drive the simulations in China, including GSWP3 (Global Soil Wetness Project Phase 3), PGF (Princeton Global meteorological Forcing), CRU-NCEP (Climatic Research Unit-National Center for Environmental

Prediction) and WFDEI (WATCH Forcing Data methodology applied to ERA-Interim reanalysis data), due to their widely applications in numerous hydrological studies (Getirana et al., 2014; Guimberteau et al., 2014, 2017, 2018; Hirschi et al., 2014; Van Den Hurk et al., 2016; Polcher et al., 2016; Schmied et al., 2016; Tangdamrongsub et al., 2018; Yang et al., 2015; Zhao et al., 2017; Zhou et al., 2018). Although they provide gridded surface climate variables at global scale, their uncertainties of representing regional climate are not clear. Through comparison of simulated SM, our study also addresses which forcing has the best performance in SM simulation in China. In fact, there is a 0.1° well-calibrated forcing data available for China (He and Yang, 2011). But the simulations driven by it are time consuming due to the resolution, which is not suitable for model validation at early stages and this forcing is not freely available on a regularly updated basis."

1.4 **"The motivation of carrying out the study specifically over China is in my view lacking a little. Also, are there no locally optimised forcing datasets available? Why run a land surface model specifically over China using global input data? Again, just make the motivation of the study a bit clearer. Why was this specific model used for the experiments, does it have any advantages specifically for China (this is actually mentioned in the model section but might be also helpful in the introduction with a little more detail)?"**

A: China is selected as the study area before global application, because (1) It covers multiple climate zones, which can help us understand different mechanisms under different climate regimes; (2) It has almost all types of anthropogenic impacts: irrigation, deforestation, afforestation, dam operations, polders, inter basin water transfer, etc, which is an ideal example to investigate climate-water-human interactions in the next step. In the introduction, we added: "Climate change strongly influences the hydrological cycle, which in turn affects ecosystems services, food security and water resources (Bonan, 2008; Piao et al., 2010; Seneviratne et al., 2010; Zhu et al., 2016). More importantly, the mechanisms of hydrological process vary across climate regimes under anthropogenic factors (Guimberteau et al., 2012; Wada et al., 2016, 2017). Covering different climate zones and most types of human activities (Rogers et al., 2016; Basheer and Elagib, 2018; Feng et al., 2016; Bouwer et al., 2009; An et al., 2017; Wu et al., 2018) China is a good test bed to investigate the hydrological complexity of climate-water-human interactions."

Yes, there is a high resolution Chinese forcing dataset but not regularly updated. Moreover, using global forcing can help us to fast extend some of the metrics developed in this study at global scale. And it will be easy to compare other parallel works using the same forcing. Related modification is shown in the reply to Comment 1.3.

To further understand the interactions between climate change, water cycle and human activities, a model that has been carefully evaluated is necessary, which integrates important mechanisms of $CO_2$, water and surface energy balances, ecological dynamics and anthropogenic processes, such as ORCHIDEE. In the introduction, the text is modified as: "Land surface models (LSMs) are able to simulate the short-term and long term SM dynamics consistently with atmospheric forcing and surface information (Rebel et al., 2012; Xia et al., 2014; Pierdicca et al., 2015) by reproducing physical processes, and

interactions with other climatic, hydrological and ecological factors (Seneviratne et al., 2010). ... In this study, the land surface model: ORCHIDEE-MICT (ORganizing Carbon and Hydrology in Dynamic EcosystEms: aMeliorated Interactions between Carbon and Temperature; Guimberteau et al. (2018)) is used to simulate SM over China (ORCHIDEE instead of ORCHIDEE-MICT for brevity). Besides land surface hydrology, ORCHIDEE simulates energy budgets and vegetation dynamics (mechanistic phenology, photosynthesis and ecosystem carbon cycling), which interact with the water cycle and climate (Guimberteau et al., 2012). Moreover, this evaluation of simulated SM controlled only by natural processes is useful to identify human effects (e.g., crops, irrigation and dam operation) on water budget in regions where there is a large misfit between model and observation."

1.5 **"Concerning the validation as a whole, the model outputs for the four experiments are compared to, in addition to in-situ measurements, GLEAM and CCI soil moisture. However, these datasets (CCI and GLEAM) should also be compared to the in-situ measurements since the mere comparison does not result in any helpful answer on which of these datasets performs any better when compared to the actual ground measurements. Both GLEAM and the CCI dataset will likely have their own problems with accurately simulating soil moisture within certain areas. At the current state of the study they are used as a kind of additional ground-truth, which they most certainly are not (and in fact, as correctly noted, GLEAM shares some of the same input data with the forcing data used for the experiments)."**

A:True. Comparison to in-situ measurement is an essential part of model validation. But this is not enough. Firstly, in-situ measurements cannot be used to estimate spatial and temporal variations of simulated SM. Secondly, because of spatial variations of climate variables and landscape, in-situ measurements can provide high accurate validation only if the atmospheric forcing is at the same spatial scale. A remote sensing product derived from multiple observations averaged or aggregated at daily time step is probably more comparable to model simulations obtained with meteorological reanalysis than local in-situ measurements.

We agree that ESA CCI and GLEAM SM should be validated before comparing to the simulations. In fact, the ESA CCI SM has been validated both at global scale (Dorigo et al., 2015) and in China (Peng et al., 2015; An et al., 2016). The GLEAM SM has been validated by ISMN as well (Martens et al., 2017). We will cite these works in the introduction of ESA CCI and GLEAM dataset. All in all, we use SM datasets for different purposes: 1) In-situ measurements are used for evaluation of fast variability, mainly decrease and recharge of top and middle soil horizons from rain events; 2) ESA CCI and GLEAM are used mainly to evaluate seasonal, inter-annual and spatial patterns of SM. This will be explained in the revised Introduction (see reply to Comment 1.2).

**2 Specific comments**

**2.1 "P2L21-32: Possibly add a sentence on soil moisture (products or raw data) data assimilation in the introduction, since the advantage / disadvantages of satellite based soil moisture products and land surface models are discussed. Data assimilation exactly tries to combine the strengths of these different types of data, such as in the GLEAM dataset."**

A: Thanks for the suggestion. A short description about data assimilation has been added in the introduction: "To overcome the uneven coverage of raw data, data assimilation is widely applied to analyze soil moisture from in-situ or satellite observations (Reichle et al., 2007; Draper et al., 2012; Martens et al., 2016). Analyzed products help us understanding SM variation and its relation to climate (Taylor et al., 2012; Liu et al., 2015b, 2017). However, to capture changes of hydrological mechanisms for future projections, measurements are not enough."

**2.2 "P3L22-29: GSWP3 is very coarse, but downsampled. Could this be especially problematic in areas within China with complex terrain?"**

A: Yes, coarse resolution is inaccurate over complex terrain regions from the Tibetan Plateau to the Sichuan Basin (Fig. 1). It is difficult to produce a high accurate forcing reanalysis dataset and simulated SM in these areas. Disagreements of simulated SM are shown in Fig. 2 and 9(a). Mismatches of meteorological variables among forcing datasets are found in these regions as well (Fig. 9(c) and 9(e)). However, according to the results, the simulated SM driven by GSWP3 is not worse than others. So in our opinion, GSWP3 is not obviously less realistic in complex terrain regions. Intercomparison should be applied among forcing datasets to further address this question, e.g. comparison with weather stations data, but it is beyond our ability and not the scope of this study.

**2.3 "P4L2-8: Is PGF still being updated? Maybe add this information to the other datasets too, or to the motivation of choosing these specific datasets."**

A: The last update of the PGF is on 13th July 2014 (http://hydrology.princeton.edu/data.pgf.php). The version information of each dataset will be added. The motivation of using these forcing date is added in the introduction. Please see our reply to Comment 1.3.

**2.4 "P4L18-22: WFDEI, why only available until 2009? Both corrected with GPCC v5 and v6?"**

A: The WFDEI_GPCC we have access to is only available until 2009 (version 31 July 2012). The version we used was only corrected with GPCC v5. The initial WFDEI forcing should be bias corrected for rainfall for use in ORCHIDEE. However, this work had not been completed when we ran the simulations.

Currently the latest WFDEI_GPCC is available. We can re-do the simulation with the latest WFDEI forcing if the reviewer strongly recommend. However, please note that there will be only one year (2010) extension due to the constraint of other forcing (e.g., GSWP3 is available only until 2010). Moreover, there is no difference of the

version we used and the latest version, except for the period length (http://www.eu-watch.org/gfx_content/documents/README-WFDEI%20(v2016).pdf).

**2.5 "P5L20: GLEAM has 0.5° resolution? I thought 0.25. It's 0.25° in Table 2, please recheck."**

A: True, it is 0.25°. Corrected.

**2.6 "P5L28 GLEAM assimilates GLDAS? I'm not so sure about this. I think it's somehow used for the background error estimation within the assimilation scheme, but please check this."**

A: We made a mistake here. The GLDAS was only used in the GLEAM data assimilation system to estimate the errors on annual basis (Martens et al., 2017). We corrected as: "... results from a combination of simulated SM from the GLEAM soil module, SMOS (the Soil Moisture Ocean Salinity satellite mission) and ESA CCI SM (ESA Climate Change Initiative Soil Moisture) through the data assimilation system developed by Martens et al. (2016). The Community Noah land surface model SM fields in GLDAS (Global Land Data Assimilation System) was used to estimate the errors of these SM products."

**2.7 "P6L26: 13 PFTs are grouped, did not understand. Only three land cover classes?"**

A: No. We used 13-PFT map including one bare soil. Each PFT has its own parameterization. PFT fractions are assigned to three soil tiles corresponding to bare soil, short vegetation (grass and crop PFTs) and forests (all tree PFTs). So each grid cell can include up to three soil tiles. The soil moisture budget of each soil tile is calculated separately, but different PFTs in the same soil tile interact as they share the same soil moisture source. In the manuscript, we modified as "Each grid cell can include up to three soil tiles: bare soil, trees and grasscrops, which are filled by the corresponding plant functional types (PFT) of the 13-PFT scheme of ORCHIDEE-MICT to allow better representation of their specific hydrology. The hydrological budget is calculated separately in each soil tile."

**2.8 "P7L10. Why aggregate results to 1 degree? This likely deteriorates the impact of 'high resolution' forcing datasets, such as WFDEI. Rather upsample coarser data by simply multiplying grid cells?"**

A: Agreed. In the revised manuscript, we will sample the simulated SM driven by PGF from 1° to 0.5° by a nearest neighbour method and re-do all analysis and plots at 0.5°. Values of metrics will be updated as well. But there is no significant impact on our results and conclusion.

**2.9 "P7L15: 'distributed to the first half of the forcing time step..' why the first half."**

A: This is the default setting of ORCHIDEE. To avoid underestimation of infiltration, precipitation amount should be assigned in the half of forcing time step. The 'first half' is the default setting of ORCHIDEE-MICT. It also can be modified as 'middle

half' or 'second half', which has no effect on hydrological simulation in principle. In the manuscript, we added: "Note that there is no effect whether the precipitation is assigned in the first or the second half time step in principle."

**2.10 "P8L1-3: Which time shift was used? Between UTC and local time (several time zones) between model and in-situ measurements. Not vital but good to know."**

A: We used UTC for all SM records. In the manuscript, we added: "In addition, the timing of all SM datasets is uniformed to the Coordinated Universal Time (UTC)."

**2.11 "P8L24: What is the exact motivation for the lag analysis? It does not seem to give any added value. How do you know one or the other are better in temporal terms? You are comparing two models."**

A: We agree that we are not able to explain which one is better by the comparison of the dynamics of SM decrease after rain events. The differences of autocorrelation can tell us the uncertainty of simulated SM from runoff, drainage and transpiration loss after a rainfall. For example, a large mismatch is found in the Yellow RB (Fig. 6h), which indicates that some unknowns existed in this region strongly influenced decrease of SM after rain events. In the paper, we discussed: "The bias of $R_k$ can be explained by higher/lower simulated evapotranspiration in YLRB/YZRB in ORCHIDEE compared to GLEAM (not shown) suggesting that the decline of ORCHIDEE $\theta_s$ is faster/slower after rainfall events than in GLEAM and lead to a lower/higher $R_k$."

Furthermore, we demonstrate that the $R_k$ curves vary among our simulations. It indicates that the autocorrelation is not only determined by the model but also by forcing data, such as precipitation intensity and frequency, which underlines our motivation: which atmospheric forcing is suitable for further hydrological study in China.

**2.12 "P9L12: Monthly values of other variables also considered ... How?"**

A: We wanted to explain that not only $P$ and $N_p$, but also other meteorological variables are inclued as indicator as well. We modified as: "We look at different climate variables to explain SM differences among simulations. ... Other meteorological indicators are incoming short/long wave radiation ($R_s/R_l$), air temperature ($T_a$), air humidity ($Q_a$) and wind speed ($W$)."

**2.13 "P9L23: These two rivers are the main ones? How much of China do these two watersheds cover? Maybe obvious for some but more background on the study region could be valuable (here and / or in the introduction)."**

A: Yes. Yangtze and Yellow are the two largest rivers in China. The watersheds of them cover 23% area of China. More importantly, the two basins cover most of agricultural and industrial regions in China. Motivation of the study area has been added in the Introduction as our reply to Comment 1.4.

More detailed explanation has been added in an extra subsection "study area" in Section 2, as: "China has multiple climate regimes, which makes hydrological simulations influenced by different variables in different regions. The land water budgets in China

is influenced by anthropogenic factors, such as irrigation (Puma and Cook, 2010), afforestation (Peng et al., 2014; Liu et al., 2015a), deforestation (Wei et al., 2018), polders (Yan et al., 2016), dams (Deng et al., 2016) and inter-basin water transfer (Li et al., 2015). Two river basins are of main interest: the Yangtze River Basin (YZRB) and the Yellow River Basin (YLRB) (red and magenta contours respectively in Fig. 1), which cover the main regions of industry and agriculture (grey regions in Fig. 1). The Yangtze River originates in the Qinghai-Tibetan Plateau and flows through two wetted traditional agricultural zones: Sichuan Basin and the plain at the downstream of the Yangtze River (Fig. 1). The Yellow River originates in the Qinghai-Tibetan Plateau as well, but it flows through another two agricultural regions (the Loess Plateau and the North China Plain) under semi-arid and semi-humid zones (Kottek et al., 2006). Our simulations cover the main part of China ([85-124°E]×[20-44°N]) including the two watersheds to assess SM dynamics not only in China but also at catchment scale."

2.14 **"This seasonality is computed across the boundaries depicted in Figure 1? A little more geographically distributed information would be helpful."**
A: Not exactly. The resolution of our simulations is coarser than the GIS data shown in Fig. 1. Thus in the analysis, the specific masks of the two river basins (not illustrated in the manuscript) do not perfectly cover the basins shown in Fig. 1. To avoid this confusion, we wrote at the end of this paragraph: "Note that in the analysis, the specific regions of the two river basins are coarser than the exact basin contours shown in Fig. 1 due to the interpolation of routing files at the resolution of our simulations."

2.15 **"P10L3: The soil moisture patterns do not necessarily match the annual mean precipitation patterns, maybe mention something about obvious monthly differences, or stronger evaporation using a specific forcing dataset? Soil moisture is not just the result of precipitation but also the other input data and model internal mechanics. No in-depth analysis is needed but some additional maps or statistics for the most important other water balance variables, e.g. evapotranspiration, or at least some sentences on the issue would be helpful. The GLEAM model you are comparing to is actually primarily developed for evapotranspiration."**
A: True. SM does not only depend on MAP. Precipitation frequency and intensity, and evapotranspiration influence SM patterns. And all of them depend on the input: atmospheric forcing. This is the third question we addressed through this study. Considering the length of the paper, Section 4.1 will be removed in the revised version, as it is not tightly related to the topic. The comparison of simulated ET to GLEAM ET has been performed and will be shortly discussed in the autocorrelation analysis.

2.16 **"'Comparison with ISMN and PKU in-situ data' seems to be a summary of the model performance for all four forcing datasets when compared to in-situ measurements. It should be noted that these are the average statistics for all carried out experiments. Instead of the next section being 'Comparison with GLEM ...' I as a reader would expect a more detailed analysis to follow (or the other way round), which now seems to be in section 4.3**

**and 4.4. Thus I would recommend to first do the in-depth comparison to in-situ measurements, followed by a comparison to other datasets thereafter. As stated at the beginning, I strongly believe that GLEAM and the CCI dataset should be validated against the in-situ measurements if you want to quantify which model actually performs better in which geographical area.”**
A: The comparison between simulated SM and in-situ measurement is a key part of validation. We will introduce the priorities of these comparisons in the Introduction (see reply to Comment 1.2). However we don't agree to move the contents in Section 4.3 and 4.4 ahead of the comparison using GLEAM and ESA CCI SM datasets. There are three research questions in this study (see Introduction). The Section 4.2 presents comparison results in order to address question: Is ORCHIDEE able to reproduce reasonable SM dynamics? The Section 4.3 is aim to demonstrate which forcing is suitable for hydrological studies in China. And the Section 4.4 is for the third research question. We think the current order is logical for readers to follow our steps to address the three questions one by one.

However, we recognize that the paragraphs of comparison with in-situ measurements were less detailed than others. In the revised manuscript, we provide a more explicit explanation to the IMSN-PKU section and reduced the GLEAM comparison and Section 4.4, as recommended by reviewer #2. Regarding the validation of ESA CCI and GLEAM SM, please check our replies to Comment 1.2 and 1.5.

**2.17 “The main finding that WFDEI performs best among forcing data is not so surprising when compared to some other studies. Again, more emphasis should be put on why this study is important specifically for China. Maybe compare the outcome of the study to other studies.”**
A: True. Motiviation of the study area has been added in the Introduction. Please check our reply to comment 1.4.

**2.18 “Table 2: Correlations are stated as being significant. Was the autocorrelation of the datasets taken into account? Also valid for the correlation at the individual stations.”**
A: No. This table does not include autocorrelation. Because the $k$-lag is an array of correlations and is difficult to show in a table. Moreover, as the answer to Comment 2.11, the autocorrelation analysis provides another aspect of the dynamics of simulated SM after rainfall. It is hard to distinguish which forcing performs better in this metrics also reflecting model processes.
We found that the initial caption of Table 2 is confusing. We checked the median $p$-value of each comparison and all of them are below 0.05. However, in a few grid cells (or stations), the $p$-value is $> 0.05$. We added the quantile of the samples with significant correlation in Table 2, and modified the caption as: “The subscripts of correlation co-efficients indicate the quantile of stations (samples) with significant correlation ($p$-value $< 0.05$).”

**2.19 “Figure 10: Should include description of variable names. Use same variable names in Figures 9 and 10.”**

A: We agree. In the revised manuscript, the variable names in Fig. 9 are replaced by their abbreviations as Fig. 10. And the description of the abbreviations is added in the caption of Fig. 9.

Table R1: Median of metrics in the comparison between ESA CCI and ORCHIDEE SM for period 1981-2009. The subscripts of correlation coefficients indicate the quantile of stations (samples) with significant correlation ($p$-value $< 0.05$).

| Dataset | Simulations | Correlation | | | RMSE ($m^3.m^{-3}$) | | |
|---------|-------------|---------|---------|--------|-------|---------|--------|
| | | China | Yangtze | Yellow | China | Yangtze | Yellow |
| | GSWP3 | $0.33_{0.96}$ | $0.29_{0.97}$ | $0.45_1$ | 0.06 | 0.06 | 0.06 |
| ESA | PGF | $0.21_{0.90}$ | $0.22_{0.95}$ | $0.24_1$ | 0.07 | 0.07 | 0.07 |
| CCI | CRU-NCEP | $0.37_{0.96}$ | $0.34_{0.97}$ | $0.46_1$ | 0.07 | 0.08 | 0.07 |
| | WFDEI | $0.47_{0.98}$ | $0.42_{0.97}$ | $0.57_1$ | 0.06 | 0.06 | 0.06 |

[Figure]

Figure R1: Number of days with available data per month of the ESA CCI soil moisture product.

[Figure]

Figure R2: Top panel: annual averaged ESA CCI soil moisture from 1979 to 2010. Bottom panel: fraction of days with available data per month in two grid cells shown in the top panel.

[revised manuscript text omitted]

---

## Author Comment (AC2) · 29 Jun 2018

**Reply to Referee #2 for "Evaluation of ORCHIDEE-MICT simulated soil moisture over China and impacts of different atmospheric forcing data" on HESSD**

Z. Yin on behalf of all co-authors

**1 Major comments**

**1.1 "The abstract contains some details that cannot be understood by reading the abstract only (it should be avoided). For instance, median R and RMSE are reported at page 1 - line 9 without mentioning with respect to which dataset they are computed. The reference to SB and LSC metrics is given but the reader is not able to understand what these metrics represent. Why are they used? Similarly for the discrepancies metric. I suggest mentioning in the abstract the results in general terms, without referring to metrics not know to the reader."**
A: True. Details of the comparison (value of metrics) are removed from the abstract. Other sentences are also slightly modified to make them more clear and brief.

**1.2 "GLEAM contains several datasets included in the atmospheric forcing datasets. It is not only ERA-Interim but also GPCC through MSWEP product. Therefore, I expect a large agreement between GLEAM and modelled soil moisture, but it does not mean the soil moisture simulations are accurate, they are simply consistent with GLEAM soil moisture (as I expected). The corresponding results should be clarified and put in perspective."**
A: Exactly. Some information is contained in both GLEAM and atmospheric forcing that we used, which may lead to a good agreement between GLEAM and simulated SM. We will discuss this issue in the revised Section of "Discussion and perspective".
However, the GLEAM SM assimilates a set of satellite observations and ground measurements (Martens et al., 2017). In addition, it has high spatio-temporal integrity in comparison to in-situ and remote sensing SM. Therefore, GLEAM is used to evaluate spatial variation of simulated SM in a long time period. In introduction we will explain the aim of using GLEAM SM, as: "Finally the GLEAM SM data (The Global Land Evaporation Amsterdam Model; Martens et al. (2017)) is compared to the simulated SM. Different from other SM datasets, GLEAM SM results from a land surface model constrained with a number of satellite and in-situ observations. This is not a direct observation but GLEAM was shown to reproduce reasonable long period SM dynamics at

global scale (Martens et al., 2017), which is valuable to evaluate ORCHIDEE simulations for both surface and root zone moisture. Furthermore, GLEAM assimilates CCI data, so that evaluation of our model against root zone moisture from GLEAM is consistent with evaluation against surface moisture from CCI. Details of the SM datasets are shown in Sect. 2.3."

1.3 **"Even though it is a satellite-based dataset (therefore, its accuracy might be not good enough), the use of ESA CCI soil moisture dataset is in my opinion good. However, why only 3 years? I agree with authors that ESA CCI soil moisture product is more accurate after 2007, but for modelling assessment, I would prefer to see a long-term comparison (1980-2017). It is highly needed and to me much more appropriate than using GLEAM."**

A: Although ESA CCI has long time coverage, data availability (the fraction of days with available measurements) is very low in China until 2006 (as shown in Fig. R1 and R2 in the reply to Reviewer #1). Moreover, the data availability varies significant in both space and time. Thus we decided to only use the data from 2007 to 2009 for comparison. In the manuscript, we modified as: "The data availability also varies along the period according to the number of instruments available and the increase of their temporal and spatial resolutions. In China, the fraction of days with available records (Figure 4 of Dorigo et al. (2015)) is lower than 20% from 1979 to 2006. More importantly, large spatial variation of gaps exists before 2006 (Fig. A1). ... To provide a reliable validation, we only use the CCI data between 2007-2009."

We suggest to present the long time comparison in online supplementary (as Fig. R3 and Table R1 shown in the reply to reviewer #1). However, if both reviewers were aware the limitation of ESA CCI already and consider the long time period comparison more important, we will present it in the revised manuscript instead of the 2007-2009 comparison.

**2 Moderate comments**

2.1 **"Too many figures, also by considering the Appendix, have been presented in the paper. I would prefer a lower number of more focused figures that would help the reader to understand clearly the main results the authors want to convey. Please try to reduce the length of the paper, mainly the results section."**

A: True. Section 4.1 will be removed. Section 4.2 (comparison between GLEAM and ORCHIDEE SM), 4.4 and 5.2 will be reduced. Figure 9 will be moved to supplementary. Figure A10, A11, A12 and A13 will be removed.

2.2 **"The sensitivity analysis linking soil moisture and meteorological variables seems to me not robust enough for being published on HESS. I might be wrong, but also the authors acknowledge this problem. I suggest removing or, at least, strongly reducing."**

A: True. The results and discussions related this analysis will be reduced in the revised version. Several related figures will be removed (see reply to the previous comment).

**3   Minor comments**

**3.1 "Acronyms and symbols should be specified the first time they appear in the text, please check."**
A: Revised. Explanation is given before the first appearance of each symbol.

**3.2 "Page 5, line 3: How is it assessed the quality of ISMN stations? Please clarify."**
A: It should be "availability". Corrected.

**3.3 "Page 5, line 17: I would not say 'only' 203 stations."**
A: True. Corrected.

**3.4 "Page 7, line 19: Soil depths are not different in the four datasets. If I am right, please remove."**
A: Here we talk about the SM datasets, not of forcing datasets or simulation outputs. Revised as: "Because the soil depths, periods and spatio-temporal resolutions are different in the four SM datasets (Sect. 2.2)..."

**3.5 "Page 10, line 16: 'an traditional' should be 'a traditional'."**
A: True. Corrected.

**3.6 "Page 10, line 20: Why the magnitude of soil moisture is systematically underestimated? Please try to find an explanation."**
A: Here we are talking about the comparison at Xuzhou station. To avoid misunderstanding, it is revised as: "However the magnitude of $\theta_t$ is systematically underestimated as well (Fig. 4)."

**3.7 "Page 12, lines 2-3: Again, why changes of precipitation regimes are not enough to predict changes in soil moisture? Please comment."**
A: It is confusing. Based on the different trends of $\theta_s$ and $P$, we infer that the trend of precipitation amount cannot well explain the trend of SM. Modified as: "The mismatch of $\theta_s$ and $P$ trends suggest that the change of precipitation amount is not the only driver of the trend of SM."

**Bibliography**

Dorigo, W. A., Gruber, A., De Jeu, R. A. M., Wagner, W., Stacke, T., Loew, A., Albergel, C., Brocca, L., Chung, D., Parinussa, R. M., and Kidd, R.: Evaluation of the ESA CCI soil moisture product using ground-based observations, Remote Sensing of Environment, 162, 380–395, https://doi.org/10.1016/j.rse.2014.07.023, 2015.

Martens, B., Miralles, D. G., Lievens, H., van der Schalie, R., de Jeu, R. A. M., Fernández-Prieto, D., Beck, H. E., Dorigo, W. A., and Verhoest, N. E. C.: GLEAM v3: satellite-based land evaporation and root-zone soil moisture, Geoscientific Model Development, 10, 1903–1925, https://doi.org/10.5194/gmd-10-1903-2017, 2017.